# A pyocin-like T6SS effector mediates bacterial competition in *Yersinia pseudotuberculosis*

Leilei Yang,[1] Shuangkai Jia,[1] Sihuai Sun,[1] Lei Wang,[1] Bobo Zhao,[1] Mengsi Zhang,[1] Yanling Yin,[1,2] Mingming Yang,[1,3] Alex M. Fulano,[4] Xihui Shen,[1,2] Junfeng Pan,[1] Yao Wang[1]

**ABSTRACT** Within the realm of Gram-negative bacteria, bacteriocins are secreted almost everywhere, and the most representative are colicin and pyocin, which are secreted by *Escherichia coli* and *Pseudomonas aeruginosa*, respectively. Signal peptides at the amino terminus of bacteriocins or ABC transporters can secrete bacteriocins, which then enter bacteria through cell membrane receptors and exert toxicity. In general, the bactericidal spectrum is usually narrow, killing only the kin or closely related species. Our previous research indicates that YPK_0952 is an effector of the third Type VI secretion system (T6SS-3) in *Yersinia pseudotuberculosis*. Next, we sought to determine its identity and characterize its toxicity. We found that YPK_0952 (a pyocin-like effector) can achieve intra-species and inter-species competitive advantages through both contact-dependent and contact-independent mechanisms mediated by the T6SS-3 while enhancing the intestinal colonization capacity of *Y. pseudotuberculosis*. We further identified YPK_0952 as a DNase dependent on $Mg^{2+}$, $Ni^{2+}$, $Mn^{2+}$, and $Co^{2+}$ bivalent metal ions, and the homologous immune protein YPK_0953 can inhibit its activity. In summary, YPK_0952 exerts toxicity by degrading nucleic acids from competing cells, and YPK_0953 prevents self-attack in *Y. pseudotuberculosis*.

**IMPORTANCE** Bacteriocins secreted by Gram-negative bacteria generally enter cells through specific interactions on the cell surface, resulting in a narrow bactericidal spectrum. First, we identified a new pyocin-like effector protein, YPK_0952, in the third Type VI secretion system (T6SS-3) of *Yersinia pseudotuberculosis*. YPK_0952 is secreted by T6SS-3 and can exert DNase activity through contact-dependent and contact-independent entry into nearby cells of the same and other species (e.g., *Escherichia coli*) to help *Y. pseudotuberculosis* to exert a competitive advantage and promote intestinal colonization. This discovery lays the foundation for an in-depth study of the different effector protein types within the T6SS and their complexity in competing interactions. At the same time, this study provides a new development for the toolbox of toxin/immune pairs for studying Gram-negative bacteriocin translocation.

**KEYWORDS** *Yersinia pseudotuberculosis*, T6SS, pyocin-like effector, DNase activity, bacterial competition

B acteria, which function as social organisms, engage in continual interactions with adjacent cells to outcompete their nonkin counterparts for vital nutrients and space (1). In this microbial competition, they have evolved an array of strategic mechanisms aimed at surpassing neighboring species, including the production of diffusible molecules, the release of phage or phage-like particles, and the use of contact-dependent antibacterial weaponry (2, 3). Among these strategies, the Type VI secretion system (T6SS) is a lethal weapon that functions as a crossbow and effectively delivers toxic effectors to target cells (4, 5). The T6SS ejects a spear-like tube carrying

Address correspondence to Yao Wang, wangyao@nwsuaf.edu.cn, or Junfeng Pan, panjf@nwsuaf.edu.cn.

The authors declare no conflict of interest.

See the funding table on p. 15.

toxic effectors outward within milliseconds, possessing sufficient force to puncture through the formidable barrier of the cell wall and two cellular membranes of Gram-negative cells (6, 7). Intriguingly, the T6SS has unveiled a myriad of effectors across a diverse spectrum of bacterial species, and antibacterial toxins are the most extensively studied among these effectors (8, 9). Notably, each antibacterial toxin is encoded in tandem with a cognate immune gene within a precise proximity, a defensive mechanism shielding the bacterium from self-intoxication or T6SS-mediated killing by neighboring sister cells (4). The bactericidal efficacy of the T6SS introduces a novel approach for modulating the composition of the intestinal and rhizosphere flora in practical applications. This discovery holds promise prospects for addressing challenges related to drug-resistant bacteria, as well as for the prevention and control of plant pathogens and environmental ecological management.

Many species have evolved to produce various antimicrobial compounds that inhibit growth of or kill rival species, encompassing an array of defenses such as broad-spectrum antibiotics, proteinaceous exotoxins, lysozymes, and bacteriocins (10). Bacteriocins, a group of potent antibacterial protein, have been identified across almost all bacterial lineages (11). Remarkably, these bacteriocins are peptides synthesized by bacterial ribosomes, closely phylogenetically related species are selected as target cells, and producers have specialized immune mechanisms to counteract the effects of these peptides (12, 13). Bacteriocins are narrow-spectrum antibacterial agents that have biological effects on closely related species, and their encoding genes are found on transposable elements, plasmids, or the producer's genome (10). The multifaceted modes of action of bacteriocins include the depolarization of lipid bilayer membranes, disruption of cell wall production (14, 15), suppression of protein synthesis (16, 17), and destruction of host nucleic acids (18). A noteworthy aspect of bacteriocins is the dependence of cellular import mechanisms on the specific target organism.

The Gram-negative bacterium *Yersinia pseudotuberculosis* is widely distributed in the natural environment, and as an intestinal pathogen, it can infect many animal species (19, 20). The genome of *Y. pseudotuberculosis* encompasses four complete sets of T6SSs that play pivotal roles in bacterial quorum sensing, host defense, and antibacterial activities (21, 22). The third Type VI secretion system (T6SS-3) gene cluster contains several effector-immune protein pairs of unknown function. Previous studies have shown that T6SS-3 secretes a novel $Ca^{2+}$- and $Mg^{2+}$-dependent DNA hydrolase effector protein, YPK_0954, which exhibits a contact-independent bactericidal function (23). Considering that YPK_0952 is also an effector protein secreted by T6SS-3, it could be inferred that YPK_0952 may possess similar antibacterial properties.

Here, YPK_0952 was identified as a pyocin-like effector, demonstrating its ability to be secreted via the T6SS-3, a mechanism crucial for its pronounced toxicity. Our results demonstrated that YPK_0952 relies on various bivalent metal ions to exert DNase activity and that this activity is inhibited by the immune protein YPK_0953. Furthermore, YPK_0952 exhibits intra-species and inter-species bactericidal activity through a T6SS-mediated contact-dependent and contact-independent mechanisms. Our data provide insight into important bacterial tactics that enable cells to establish a foothold in competitive niches, not only through the secretion of specialized metabolites and bacteriocins that antagonize microbial competitors but also through contact-dependent and contact-independent suppression of rival bacteria by toxins. The cross-species bactericidal action of pyocin-like effectors released by the T6SS in *Y. pseudotuberculosis* makes bacteriocins appealing therapeutic agents, particularly in the fight against multidrug-resistant diseases.

## MATERIALS AND METHODS

### Bacterial strains and growth conditions

The bacterial strains and plasmids used in this study are listed in Table S1. *Y. pseudotuberculosis* strains were grown in Yersinia-Luria-Bertani (YLB) broth (1% tryptone, 0.5% yeast extract, 0.5% NaCl) or M9 medium (6 g L$^{-1}$ Na$_2$HPO$_4$, 3 g L$^{-1}$ KH$_2$PO$_4$, 0.5 g L$^{-1}$ NaCl, 1 g L$^{-1}$ NH$_4$Cl, 1 mM MgSO$_4$, 0.1 mM CaCl$_2$, 0.2% glucose) at 30°C or 26°C with suitable antibiotics as necessary. At 37°C, the *Escherichia coli* strains were grown in Luria-Bertani (LB) media supplemented with suitable antibiotics. When necessary, antibiotics were added at the following concentrations: nalidixic acid, 20 µg mL$^{-1}$; ampicillin, 100 µg mL$^{-1}$; kanamycin, 50 µg mL$^{-1}$; chloramphenicol, 20 µg mL$^{-1}$.

### Plasmid construction

The primers used in this study are listed in Table S2. To generate expression plasmids, the gene encoding YPK_0953 was amplified by PCR, and the resulting DNA fragment was digested and subsequently introduced into linearized pGEX6p-1 and pET28a, leading to the corresponding plasmid derivatives. Using the same methodology, the expression clones of YPK_0952 and YPK_0952-YPK_0953 were obtained. The QuickMutation Site-Directed Mutagenesis Kit (Beyotime Biotechnology, China) was used for site-directed mutagenesis of pET28a-*ypk_0952*$^{F441A}$ and pET28a-*ypk_0952*$^{Y450A}$. In brief, we designed site-mutated specific primers *ypk_0952*$^{F441A}$-F/*ypk_0952*$^{F441A}$-R or *ypk_0952*$^{Y450A}$-F/*ypk_0952*$^{Y450A}$-R, according to the manufacturer's instructions, pET28a-*ypk_0952* was used as the template, and pET28a-*ypk_0952*$^{F441A}$ was amplified by PCR using the primers *ypk_0952*$^{F441A}$-F/*ypk_0952*$^{F441A}$-R. Following the PCR, 1 µL of DpnI was added to the PCR mixture, thoroughly mixed, and then incubated at 37°C for 5 min. The resulting DpnI-digested product was used for transformation, and the correct plasmid, pET28a-*ypk_0952*$^{F441A}$, was identified through sequencing. Similarly, the point mutation plasmid, pET28a-*ypk_0952*$^{Y450A}$, was constructed. To construct plasmids utilized in bacterial two-hybrid complementation assays, the *ypk_0952* or *ypk_0953* genes were PCR amplified from *Y. pseudotuberculosis* genomic DNA using appropriate primers. The resulting DNA fragments were subsequently cloned and inserted into the corresponding sites of the pKT25 and pUT18C vectors. To create the pDM4-Δ*ypk_0952* knockout plasmid, PCR was conducted to generate upstream and downstream fragments flanking *ypk_0952* using the primer pair *ypk_0952*-M1F-*Sph*I/*ypk_0952*-M1R and *ypk_0952*-M2F/*ypk_0952*-M2R-*Sal*I. Subsequently, the upstream and downstream fragments were ligated through overlap PCR with the primer pair *ypk_0952*-M1F-*Sph*I/*ypk_0952*-M2R-*Sal*I. The resulting fragment, along with plasmid pDM4, was digested with *Sph*I and *Sal*I (TaKaRa, Dalian, China) and was subsequently ligated using DNA ligase (TaKaRa, Dalian, China) to yield plasmid pDM4-Δ*ypk_0952*. The pDM4-Δ*ypk_0952*Δ*ypk_0953* knockout plasmid was constructed in a similar manner using the primers listed in Table S2. To complement the Δ*ypk_0952* mutant, the primers *ypk_0952*-F-*Pst*I and *ypk_0952*-R-*Bam*HI were used to amplify the *ypk_0952* gene fragment from *Y. pseudotuberculosis* genomic DNA. PCR-amplified *ypk_0952* was cloned and inserted into the *Pst*I and *Bam*HI sites of the plasmid pKT100 to produce pKT100-*ypk_0952*. A similar procedure was followed for the construction of the complementary plasmid pKT100-*ypk_0953*. The integrity of the inserts in all the constructs was confirmed through DNA sequencing.

### In-frame deletion and complementation

To generate in-frame deletion mutants, we harnessed *E. coli* S17-1 λ-pir-mediated conjugational mating to transfer pDM4 derivatives into *Y. pseudotuberculosis*. Briefly, 700 µL of *E. coli* S17-1 containing the pDM4 derivative was mixed with 300 µL of *Y. pseudotuberculosis* and was concentrated to 50 µL for use. Subsequently, we placed the mixture on a nonselective LB plate and incubated it for 16 h at 30°C to promote mating. Plasmid integration into the *Y. pseudotuberculosis* chromosome by single crossover events was carried out using YLB plates with 20 µg mL$^{-1}$ chloramphenicol and 20 µg mL$^{-1}$ nalidixic

acid. The resulting chloramphenicol-resistant colonies were cultivated overnight in LB broth to facilitate a second crossover. Subsequent selection for the loss of the genome-integrated *sacB*-containing plasmid was performed on YLB plates supplemented with 20% sucrose and 20 µg mL$^{-1}$ nalidixic acid. Colonies thriving on this selective medium were evaluated for chloramphenicol sensitivity through parallel spotting on YLB plates containing chloramphenicol or nalidixic acid. Deletions were confirmed by PCR and DNA sequencing (24, 25). For complementation in relevant *Y. pseudotuberculosis* strains, the pKT100 derivatives were transformed into relevant *Y. pseudotuberculosis* strains by electroporation.

## Overexpression and purification of recombinant protein

To express and purify His$_6$ and GST-tagged recombinant proteins, plasmid derivatives of pET28a and pGEX6p-1 were introduced into competent *E. coli* BL21 (DE3). The bacterial culture was cultivated at 37°C in LB medium until the optical density at 600 nm (OD$_{600}$) reached 0.4, after which the temperature was reduced to 22°C before the induction phase, at which point 0.25 mM isopropyl-beta-D-thiogalactopyranoside (IPTG) was introduced, and the cultivation temperature was maintained at 22°C for an additional 10 h. Harvested cells were disrupted by sonification and were purified with His•Bind Ni-NTA resin (Novagen, Madison, WI, USA) or GST•Bind Resin (Novagen, Madison, WI, USA) according to the instructions provided by the manufacturer. The eluted recombinant proteins were dialyzed against buffer (137 mM NaCl, 2.7 mM KCl, 10 mM Na$_2$HPO$_4$, 2 mM KH$_2$PO$_4$, and 10% glycerol, pH 7.4) at 4°C. Subsequently, the proteins were stored at −80°C until further use. Protein concentrations were determined using the Bradford assay (26). The analysis of recombinant proteins was conducted utilizing SDS-PAGE.

## Western blot analysis

The protein samples were resolved by SDS-PAGE and were transferred onto polyvinylidene fluoride membranes (Millipore, Burlington, MA, USA) before they were blocked in 5% (wt/vol) bovine serum albumin (BSA) for 8 h at 4°C. The membrane was then incubated overnight at the same temperature with the following primary antibodies: anti-His (Santa Cruz Biotechnology, catalog no. sc-8036, lot number: I1018), 1:1,000; and anti-GST (Santa Cruz Biotechnology, catalog no. sc-53909, lot number: F2413), 1:1,000. Each membrane was subjected to five successive washes in Tris buffered saline with tween 20 (TBST) (50 mM Tris-HCl, 150 mM NaCl, 0.05% Tween 20, pH 7.4) and then incubated with 1:10,000 diluted horseradish peroxidase-conjugated secondary antibodies (Shanghai Genomics, catalog no. DY60203, lot number: 20614) for 4 h at 4°C, followed by an additional five washes with TBST buffer. Signals were detected using an ECL Plus kit (GE Healthcare, Piscataway, NJ, USA) and a chemiluminescence imager (Tanon 5200Multi, Beijing, China).

## GST pull-down assay

The glutathione S-transferase (GST) pull-down assay was conducted as previously described, with minor modifications (27, 28). To analyze protein interactions, purified GST fusion protein was mixed with 6× His fusion protein (final concentration of 2.5 µM each) in Tris-EDTA-NaCl (TEN) buffer (100 mM Tris-HCl, 10 mM EDTA, 100 mM NaCl, pH 8.0) on a rotator for 2 h at 4°C. Then, a mixture of 40 µL of sample and 10 µL of 5× SDS-PAGE loading buffer was used as the input. GST or the irrelevant protein YPK_3549 fused to GST was used as a negative control. After adding 40 µL of prewashed glutathione bead slurry, binding proceeded for another 2 h at 4°C. The beads were subsequently washed five times with TEN washing buffer (100 mM Tris-HCl, 10 mM EDTA, 500 mM NaCl, pH 8.0) for 15 min each. An equal volume of glutathione beads was transferred to a 1.5-mL eppendorf (EP) tube, the supernatant was discarded, and the glutathione beads were resuspended in 50 µL of 1× SDS-PAGE loading buffer. Retained proteins were resolved by SDS-PAGE and visualized by Western blot analysis.

## Bacterial two-hybrid assay

To conduct bacterial two-hybrid complementation assays (29, 30), we co-transformed derivatives of pKT25 and pUT18C with *E. coli* BTH101, followed by cultivation on a MacConkey plate (ampicillin 100 µg mL$^{-1}$, kanamycin 50 µg mL$^{-1}$, IPTG 1 mM) at 30°C. Concurrently, we co-transformed the pKT25-zip/pUT18C-zip and pKT25/pUT18C plasmids with *E. coli* BTH101, which served as positive and negative controls, respectively. Interactions were evaluated on MacConkey medium, where a red colony signified a protein interaction, and a white colony signified the absence of interaction. To quantify the interaction efficiencies between diverse proteins, we gauged β-galactosidase activities in liquid cultures. In summary, overnight cultures, diluted to 1%, were grown in LB broth supplemented with antibiotics at 30°C until they reached an OD$_{600}$ of 1.0. The OD$_{600}$ of each sample was measured, and 50 µL of the culture was transferred to a 1.5-mL EP tube. In a fume hood, each culture was treated as follows. Briefly, 420 µL of Z-buffer, 20 µL of chloroform, and 10 µL of 0.1% SDS were added to each EP tube. The mixture was incubated at 30°C for 30 min to lyse the cells. Moreover, the substrate was prepared by dissolving 0.004 g of *o*-nitrophenyl-β-galactoside (ONPG) in 1 mL of freshly prepared substrate solution and incubating it at 30°C, and 220 rpm/min for 30 min. Subsequently, β-galactosidase activities were evaluated using ONPG as the substrate. When 100 µL of the substrate was added to the first treated culture, the mixture was mixed well, and once a significant color change occurred, the reaction was stopped by the addition of 250 µL of 1 M Na$_2$CO$_3$. The wash time was stopped, and the precipitate was allowed to settle for 3–5 min. Then, 200 µL of the supernatant was transferred to a microplate, and OD$_{420}$ and OD$_{500}$ were measured. The method for calculating enzyme activity is as follows:

$$\text{Units} = \frac{1,000 \times (\text{OD}_{420} - 1.75 \times \text{OD}_{500})}{\text{T} \times \text{OD}_{600} \times \text{V}}$$

T, reaction time (minutes); V, volume of the bacterial culture (milliliter)

## Bioinformatics analysis

All *Y. pseudotuberculosis* genomic data were sourced from the Kyoto Encyclopedia of Genes and Genomes (KEGG) Database (http://www.kegg.jp). The protein sequences associated with pyocin across various bacterial species were obtained from the National Center for Biotechnology Information (NCBI) Protein Database (https://www.ncbi.nlm.nih.gov/protein). Multiple sequence alignments were performed using CLUSTAL W, and a phylogenetic tree was generated with MEGA 11. The structural model was simulated using SWISS-MODEL (http://www.swissmodel.org) (31).

## *In vitro* DNase assay

The DNase assay was performed as previously described (23). Purified His$_6$-YPK_0952 protein (0.1 µM) was incubated with λ DNA (0.35 µg, Takara, Japan, catalog no. 3010) in reaction buffer (20 mM MES, 100 mM NaCl, pH 6.9). In all, 4 mM EDTA or 2 mM other divalent metals or other components were added to the reaction system as indicated in different experiments. DNA hydrolysis was carried out at 37°C for 30 min, and the integrity of the DNA was analyzed by 0.7% agarose gel electrophoresis.

To analyze the ability of *Y. pseudotuberculosis* to degrade DNA, *Y. pseudotuberculosis* strains were cultured overnight in YLB at 30°C, transferred into 5 mL M9 medium, and grown at 26°C until the OD$_{600}$ reached 0.55. The culture was centrifuged at 5,000 rpm/min for 20 min to collect the supernatant, which was then further centrifuged at 9,900 rpm/min for 50 min. The resulting supernatant was sterile filtered through a 0.22-µm filter. The supernatants containing toxins were subjected to DNase activity assays as described above. The wild-type (WT), Δ*clpV3*, and Δ*ypk_0952* supernatants (13 µL) were mixed with λ DNA (0.35 µg, Takara, Japan, catalog no. 3010) in reaction buffer

(20 mM MES, 100 mM NaCl, 2 mM $CoCl_2$, pH 6.9) and were incubated at 37°C for 1 h or 2 h. The integrity of the DNA was analyzed by 0.7% agarose gel electrophoresis.

## Fluorescence microscopy of *E. coli* cells and staining with DAPI

4′,6-diamidino-2-phenylindole (DAPI) staining was performed following the previously described method, with slight modifications (4). Overnight cultures of *E. coli* DH5α were washed with M9 medium, diluted 40-fold, and then transferred to 5 mL sterile supernatants of WT, Δ*clpV3*, and Δ*ypk_0952* strains, respectively. The cultures were then incubated at 37°C for 4 h. The bacterial cells were collected, washed with phosphate-buffered saline (PBS), fixed for 5 min, and subsequently stained with 10 µg/mL DAPI in PBS containing 0.3% Triton X-100 for 30 min at 37°C (Solarbio, China). After washing three times with PBS, the samples were examined using confocal laser scanning microscopy (Stellaris 8, Germany).

## Antibacterial assay

The antibacterial assay was conducted as described, with minor modifications (23). Overnight cultures of *E. coli* DH5α were washed with M9 medium, diluted 40-fold, and then transferred to 5 mL sterile supernatants of WT, Δ*clpV3*, and Δ*ypk_0952* strains, respectively. The cultures were then incubated at 37°C for 4 h. After treatment, the cultures were serially diluted and plated onto LB agar plates. The plates were then incubated at 37°C for 12 h, and the number of colony-forming unit (CFU) was counted to determine the quantity of the DH5α strain.

## Terminal deoxynucleotidyl transferase dUTP nick-end labeling (TUNEL) and flow cytometry analysis

Overnight cultures of *E. coli* BL21 (DE3) containing the pET28a plasmid or its derivatives expressing YPK_0952 alone (pET28a-*ypk_0952*) or YPK_0952^F441A alone (pET28a-*ypk_0952*^F441A^), YPK_0952^Y450A alone (pET28a-*ypk_0952*^Y450A^), or YPK_0952-YPK_0953 together (pET28a-*ypk_0952-ypk_0953*) were diluted 100-fold in LB broth and were incubated at 26°C with shaking at 180 rpm/min shaking for 2 h. After incubation, the expression of toxins and immunity genes were induced by adding 0.5 mM IPTG, and cultivation was continued for an additional 4 h at 26°C. The collected cells were washed with PBS, fixed with 4% paraformaldehyde fix solution (Beyotime Biotechnology, China), incubated for 5 min in PBS supplemented with 0.3% Triton X-100, and stained using a one-step TUNEL cell apoptosis detection kit (Beyotime Biotechnology, China). When the genomic DNA broke, the exposed 3′-OH ends were labeled with the green fluorescent probe fluorescein isothiocyanate (FITC) catalyzed by terminal deoxynucleotidyl transferase (TdT). This labeling was detected by flow cytometry (CytoFLEX, Beckman). FlowJo_V10.8.1 was used to analyze ten thousand cells collected from each sample (8).

## Growth inhibition assay

*Y. pseudotuberculosis* strains harboring the WT (pKT100), Δ*ypk_0952*Δ*ypk_0953* (pKT100), Δ*ypk_0952*Δ*ypk_0953* (pKT100-*ypk_0952*), and Δ*ypk_0952*Δ*ypk_0953* (pKT100-*ypk_0953*) strains were grown to a stable stage in YLB media. The steady-stage cultures were adjusted to the same $OD_{600}$ value and were diluted 100-fold in YLB broth containing appropriate antibiotics. The plates were incubated at 30°C with continuous monitoring of culture growth through $OD_{600}$ measurements at 2-h intervals. Alternatively, cultures were adjusted to the same $OD_{600}$ value during the stable phase and then serially diluted 10-fold. Subsequently, 2.5 µL of the bacterial suspension from each dilution was inoculated onto antibiotic media, and the cultures were incubated at 30°C.

*E. coli* BL21 (DE3) harboring the pET28a empty vector, pET28a-*ypk_0952*, pET28a-*ypk_0952*^F441A^, pET28a-*ypk_0952*^Y450A^, pET28a-*ypk_0952-ypk_0953*, pET28a-*ypk_0952*^Δpyocin_S^, and pET28a-*ypk_0952*^pyocin_S^ were grown in LB media. Overnight cultures were adjusted to the same $OD_{600}$ value and were diluted 100-fold into LB broth containing appropriate

antibiotics. Following a 2-h incubation at 26°C and 180 rpm/min, the expression of the recombinant proteins was induced with 0.5 mM IPTG. The cultures were continually incubated under the same conditions, and growth was monitored by measuring the $OD_{600}$ at 2-h intervals. Alternatively, after adjusting the stationary-phase cultures to the same $OD_{600}$ value, they were continuously diluted 10-fold. Subsequently, 2.5 µL of bacterial suspension from each concentration gradient was spotted on a plate containing $Km^R$ and 0.3 mM IPTG, and the plate was cultured at 26°C (23).

## Intra-species and inter-species competition *in vitro*

For intra-species competition assays (32), strains cultured overnight were normalized to $OD_{600}$ = 1.0 in M9 media and were mixed at a 1:1 predator:prey ratio. The resultant mixtures (10 µL) were placed onto a 0.22-µm nitrocellulose membrane (Nalgene) on M9 agar plates, which were allowed to interact at 26°C for 48 h (for contact-dependent competition). For intra-species contact-independent competition performed on a solid surface, 5 µL of prey strain was first spotted onto a 0.22-µm nitrocellulose membrane on M9 agar plates. After the liquid dried, another layer of 0.22-µm nitrocellulose membrane was placed on top, and 5 µL of predator strain was spotted at the same position on the second membrane. The plates were then incubated at 26°C for 48 h. To facilitate subsequent screening on YLB plates, the predator and prey strains were distinguished through the labels pACYC184 ($Cm^R$) and pKT100 ($Km^R$), respectively. At the indicated time points after the competition, the CFU of the prey strains was ascertained through plate counts.

For inter-species competition assays (23), overnight-grown *Y. pseudotuberculosis* strains harboring pKT100 ($Km^R$) and *E. coli* DH5α strains harboring pACYC184 ($Cm^R$) were washed three times with M9 medium and were adjusted to $OD_{600}$ = 1.0. The *Y. pseudotuberculosis* strains were diluted 10-fold, and the target strains were attenuated 100-fold and were mixed so that the ratio of predator to prey strains was 10 to 1 in M9 liquid. The cocultures were spotted onto a 0.22-µm nitrocellulose membrane (Nalgene) placed on M9 agar plates at 26°C for 12 h (for contact-dependent competition). The same method as used for intra-species competition was employed to conduct contact-independent competition on a solid surface; 5 µL of prey strain was first spotted onto a 0.22-µm nitrocellulose membrane on M9 agar plates. After the liquid dried, another layer of 0.22-µm nitrocellulose membrane was placed on top, and 5 µL of predator strain was spotted at the same position on the second membrane. The plates were then incubated at 26°C for 24 h. Following the competition, the mixture was subjected to plate counting to determine the number of DH5α strains.

## Murine colonization assay

Female 6-week-old BALB/c mice were acclimated in the laboratory for 3 d. Mice were orally gavaged with $10^9$ CFUs of the indicated *Y. pseudotuberculosis* strains and were monitored for 24 h. At the end of the experiment, six mice from each group were euthanized, and colon tissues (from the WT group: 0.43, 0.29, 0.39, 0.34, 0.35, 0.36 g; from the Δ*ypk_0952*Δ*ypk_0953* group: 0.32, 0.51, 0.39, 0.33, 0.33, 0.36 g) and small intestine tissues (from the WT group: 0.95, 1.07, 0.85, 0.9, 0.81, 0.76 g; from the Δ*ypk_0952*Δ*ypk_0953* group: 0.84, 0.96, 1.14, 0.86, 1.49, 1.1 g) were weighed. The colon and small intestine tissues were then homogenized in 0.5 mL of PBS on ice, and samples of different dilutions were plated on selective YLB agar containing appropriate antibiotics for CFU counting (23). This was followed by subsequent absolute quantification of CFU by normalization of each sample to the initial pellet weight.

## Statistical analysis

All experiments were repeated at least three times with similar results. Statistical analyses were performed using GraphPad Prism 9.0 software. Statistical analyses of colonization in mice were performed using a two-sided Mann–Whitney test. The growth inhibition assay

and the median fluorescence intensity were analyzed using ordinary one-way analysis of variance (ANOVA) with Tukey's multiple comparison test. All other experiments were analyzed using unpaired, two-tailed Student's *t*-tests. *$P < 0.0332$, **$P < 0.0021$, ***$P < 0.0002$, ****$P < 0.0001$.

## RESULTS

### Identification of S-type pyocin-like T6SS effectors

In the previous investigation, we meticulously pinpointed a gene locus, YPK_0952-YPK_0958 within the *Y. pseudotuberculosis* genome, which encodes numerous T6SS effector-immunity pairs (Fig. S1). Within this array of effectors, YPK_0954, also known as Tce1, stands out as a nuclease toxin responsible for facilitating contact-independent T6SS antagonism. The primary open reading frame (ORF), YPK_0952, is distinguished by a typical PAAR domain at its N-terminus and an S-type pyocin domain at its C-terminus. YPK_0952 predominantly functions as a T6SS effector secreted by T6SS-3 (23). Multiple sequence alignment revealed that the Pyocin_S domain in YPK_0952 was similar to those *Pectobacterium parmentieri* and *Pectobacterium atrosepticum* (Fig. 1A). An exploration of the phylogenetic relationships revealed a close association between YPK_0952 of *Y. pseudotuberculosis* and S-type pyocins of *P. parmentieri* and *P. atrosepticum* (Fig. 1B). Using SWISS-MODEL, we derived structural models for YPK_0952 and *P. parmentieri* SCC3193, an S-type pyocin, as illustrated in Fig. 1C. According to these bioinformatic analyses, the *ypk_0952* gene within the *Y. pseudotuberculosis* genome is designated a pyocin-like gene.

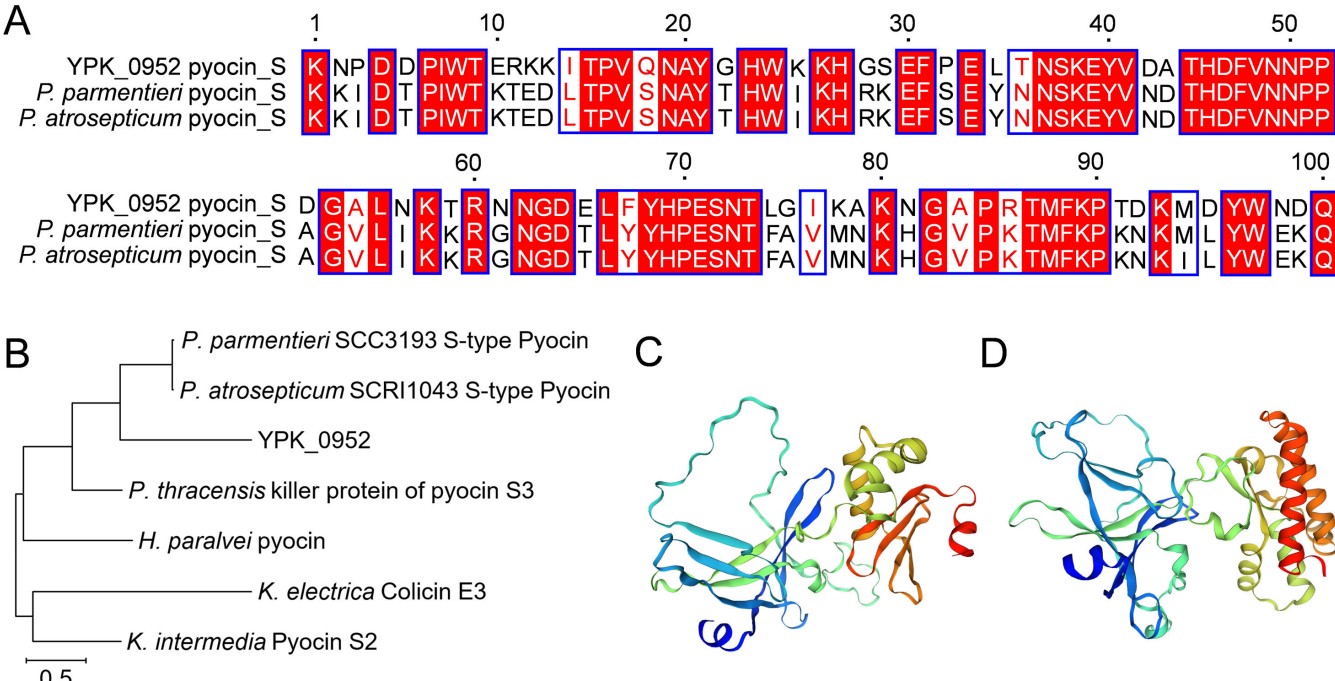

**FIG 1** Identification of S-type pyocin-like T6SS effector. (A)Multiple sequence alignment of YPK_0952 with pyocin_S from *P. parmentieri* and *P. atrosepticum*. The multiple sequence alignment was performed using CLUSTAL W, and the figure was produced by ESPript 3.0. The conserved sequences were colored in red padding, and the less conserved ones in white. (B) Phylogenetic relationship of *Y. pseudotuberculosis* YPK_0952 with pyocin in different bacteria. Different protein sequences were obtained from the NCBI Protein Database. Accession numbers are as follows: YPK_0952 (WP_011192914.1); *P. parmentieri* SCC3193 S-type Pyocin (MCL6354409.1); *P. atrosepticum* SCRI1043 S-type Pyocin (CAG74573.1); *Photorhabdus thracensis* killer protein of pyocin S3 (WP_046976409.1); *Hafnia paralvei* pyocin (WP_061059990.1); *Klebsiella electrica* Colicin-E3 (QDI06789.1); *Kluyvera intermedia* Pyocin S2 (VDZ82953.1). The phylogenetic tree was constructed using MEGA 11 by the neighbor-joining method, and multiple sequence alignment was performed using CLUSTAL W. The scale bar indicates the percentage of divergence (distance). (C and D) The structure model of YPK_0952 and *P. parmentieri* pyocin_S were simulated using SWISS-MODEL. The structure is shown as a cartoon in rainbow colors, from the N-terminus in blue to the C-terminus in red.

## YPK_0952 is toxic to *Y. pseudotuberculosis* and *E. coli*

To validate the deleterious potential of YPK_0952, we conducted toxicity assays in both *Y. pseudotuberculosis* and *E. coli*. Subsequently, we performed a complementation analysis of the Δ*ypk_0952*Δ*ypk_0953* mutant by introducing either pKT100-*ypk_0952*, pKT100-*ypk_0953*, or an empty vector to discern the nuances of its toxicity. Intriguingly, the Δ*ypk_0952*Δ*ypk_0953* mutant harboring pKT100-*ypk_0952* exhibited pronounced growth stagnation, while no discernible growth variation was observed in the mutants carrying the empty vector or pKT100-*ypk_0953* in comparison to the WT, underscoring the inherent toxicity of YPK_0952 in the milieu of *Y. pseudotuberculosis* (Fig. 2A and B).

The expression of YPK_0952 in *E. coli* resulted in significant growth inhibition. To further understand the underlying factors, we examined the toxicity profiles of distinct protein interceptors associated with YPK_0952. The results showed that the YPK_0952 carboxyl-terminal interceptor (YPK_0952[pyocin_S]) was more toxic than the full-length YPK_0952 in *E. coli* BL21 (DE3) (Fig. 2C and D). Notably, this growth inhibition was alleviated upon coexpression of the immediate downstream gene YPK_0953, within the *ypk_0952-ypk_0953* bicistron. This discerning outcome suggests that YPK_0953 is the putative cognate immune protein for YPK_0952. By performing site-directed mutagenesis toxicity screening on multiple highly conserved sites within the carboxyl-terminal toxic region (Fig. S2), we found that YPK_0952[F441A] and YPK_0952[Y450A] did not inhibit the growth of *E. coli* BL21 (DE3) (Fig. 2E and F), so the above two sites are toxicity-critical amino acids.

## Direct binding between YPK_0952 and YPK_0953

To investigate the mechanism underlying the immunity conferred by YPK_0953, we conducted rigorous glutathione S-transferase (GST) pull-down and bacterial two-hybrid assays. The results distinctly revealed specific interactions between YPK_0952 and YPK_0953 (Fig. 3A and B). These results underscore that YPK_0952 functions as a T6SS-3-secreted antibacterial effector and that its inherent toxicity is effectively neutralized by the YPK_0953 immune protein.

## YPK_0952 exhibits DNase activity

In the exploration of YPK_0952 as a T6SS effector, we investigated the biochemical activities of this protein. Notably, YPK_0952 harbors an S-type pyocin domain in the C-terminus of the protein, suggesting its potential as a nuclease toxin. With purified YPK_0952 (Fig. S3A), incubation with λ DNA in the same reaction buffer as that used for DNase I resulted in a dramatic DNA degradation pattern akin to the effects observed with DNase I. At the same time, the addition of excess EDTA served as a potent inhibitor of both YPK_0952 and DNase I activities (Fig. 4A). The catalytic activity of YPK_0952 as a DNase was critically contingent on the presence of $Mg^{2+}$, $Ni^{2+}$, $Mn^{2+}$, and $Co^{2+}$ (Fig. 4B). Thus, YPK_0952 is an endonuclease that cleaves DNA. Consistent with its proposed role as an immune protein for YPK_0952, the addition of YPK_0953 to the reaction mixture effectively decreased the DNase activity of YPK_0952 (Fig. 4C). Furthermore, the supernatant of *Y. pseudotuberculosis* WT showed the best degradation effect on DNA, while the supernatants of Δ*clpV3* and Δ*ypk_0952* exhibited almost no degradation of DNA (Fig. S6). YPK_0952[F441A] and YPK_0952[Y450A] mutations cause YPK_0952 to lose its DNase activity, further proving that these two sites are critical amino acids for toxicity (Fig. 4D). Purification of the mutant protein (Fig. S3B and C) revealed its inability to cleave DNA, distinctly differing from that of the WT YPK_0952 protein (Fig. 4D). This unequivocally substantiates that the DNase activity attributed to YPK_0952 is not a consequence of contamination.

Further confirmation of the *in vivo* DNase activity of YPK_0952 was achieved through the TUNEL assay. *E. coli* cells expressing YPK_0952 displayed prominent TUNEL signals, indicating DNA fragmentation. Conversely, *E. coli* expressing YPK_0952[F441A] and YPK_0952[Y450A] and those coexpressing YPK_0952-YPK_0953 remained unlabeled, akin to

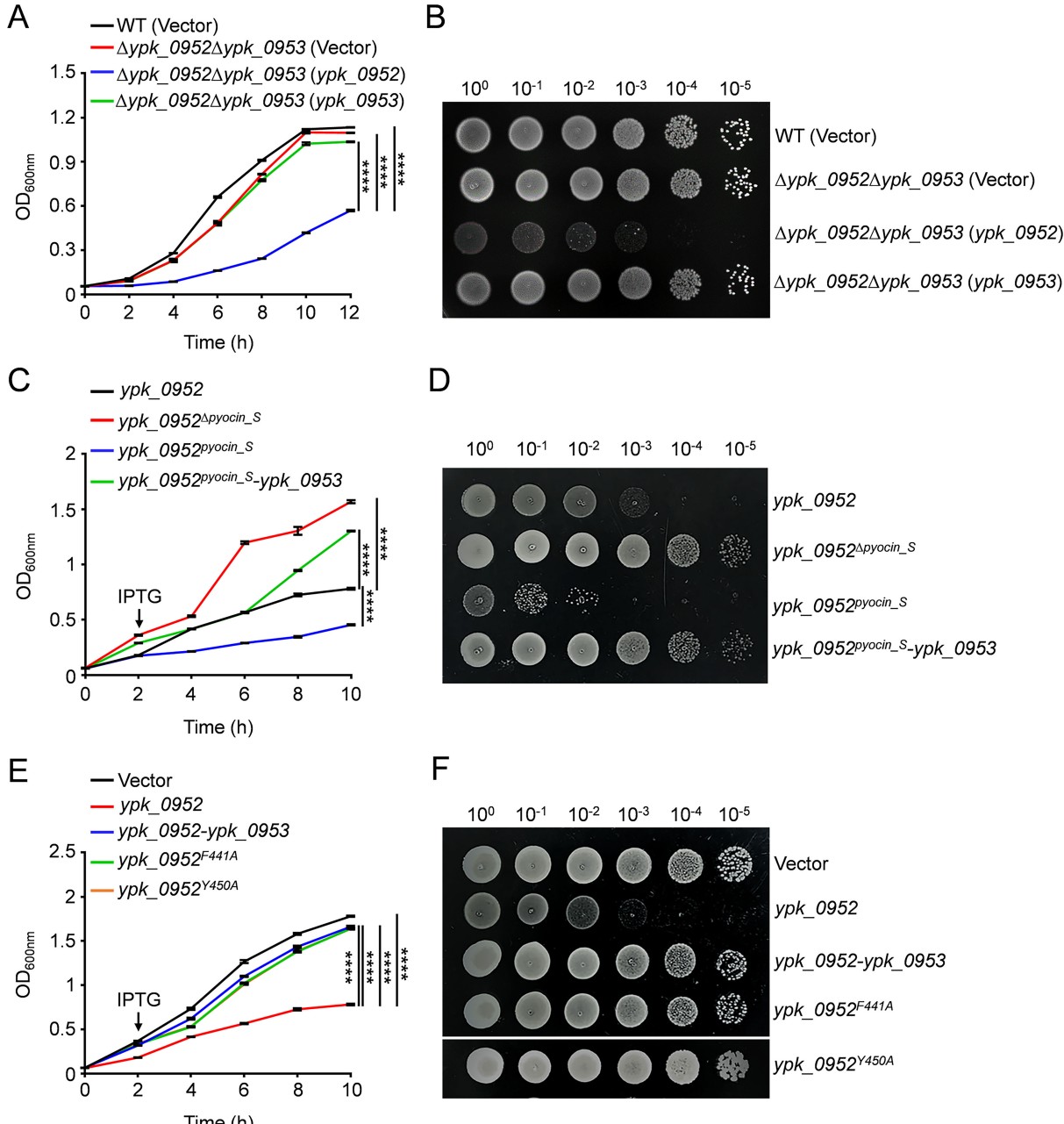

**FIG 2** YPK_0952 is toxic in *Y. pseudotuberculosis* and *E. coli*. (A) Growth curves of *Y. pseudotuberculosis* harboring indicated plasmids were obtained by measuring the $OD_{600}$ at 2-h intervals. Ordinary one-way ANOVA with Tukey's multiple comparison test with Δ*ypk_0952*Δ*ypk_0953* (*ypk_0952*). (B) Growth of *Y. pseudotuberculosis* harboring indicated plasmids on YLB solid medium. Shown from left to right are increasing serial 10-fold dilutions. (C and E) Growth curves of *E. coli* BL21 (DE3) harboring indicated plasmids were obtained by measuring the $OD_{600}$ at 2-h intervals. Ordinary one-way ANOVA with Tukey's multiple comparison test with *ypk_0952*. (D and F) Growth of *E. coil* BL21 (DE3) harboring plasmids with the indicated genes, or empty plasmids, under inducing (IPTG) conditions. Shown from left to right are increasing serial 10-fold dilutions. Ordinary one-way ANOVA with Tukey's multiple comparison test with Vector. Error bars represent the mean ± standard deviation (SD) of three independent experiments. *$P < 0.0332$, **$P < 0.0021$, ***$P < 0.0002$, ****$P < 0.0001$. ns, not significant.

the vector-only control (Fig. 4E; Fig. S5). These findings indicate that YPK_0952 can be used as a bona fide DNase.

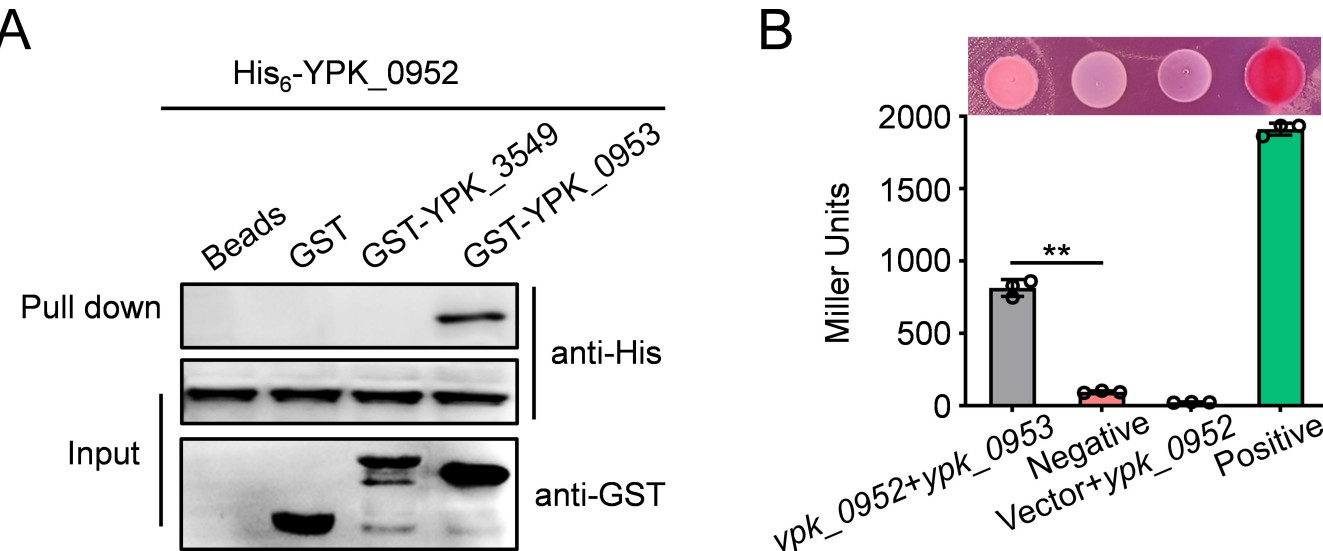

**FIG 3** Direct binding between YPK_0952 and YPK_0953 was detected using GST pull-down and bacterial two-hybrid assays. (A) His$_6$-YPK_0952 was incubated with GST-YPK_0953, GST, or an irrelevant recombinant protein GST-YPK_3549, and the protein complexes captured on glutathione beads were detected using Western blotting. (B) Interactions were assessed using MacConkey maltose plates (upper) and the β-galactosidase assay (lower). Error bars represent the mean ± SD of three independent experiments, with two-tailed, unpaired Student's $t$-test. *$P < 0.0332$, **$P < 0.0021$, ***$P < 0.0002$, ****$P < 0.0001$.

## YPK_0952 mediates contact-dependent and contact-independent T6SS killing

By scrutinizing the impact of YPK_0952-YPK_0953, an effector-immunity pair, on bacterial antagonism, we conducted growth competition assays employing labeled derivatives of *Y. pseudotuberculosis*. Co-culture under conditions conducive to cell contact. Intriguingly, the WT predator displayed a sixfold growth advantage when pitted against the Δ*ypk_0952*Δ*ypk_0953* prey. However, this growth advantage was nullified upon introducing YPK_0953 into the prey milieu. Notably, the YPK_0952-mediated growth advantage is contingent upon a fully operational T6SS-3, as evidenced by the complete eradication of this advantage following the deletion of *clpV3* (Fig. 5A). Further analysis of contact-independent competition experiments revealed that YPK_0952 could still mediate the bactericidal function of T6SS when prey and predator bacteria were isolated using a 0.22-μm nitrocellulose membrane (Fig. S7A). It was shown that *Y. pseudotuberculosis* T6SS-3 utilizes YPK_0952 for both contact-dependent and contact-independent bacterial competition.

The extant study posits that pyocins only typically zero in on and exterminate bacteria of the same or closely related species (33). Conversely, the expression of YPK_0952 in *E. coli* manifested overt toxicity. To further investigate YPK_0952-mediated contact-dependent and contact-independent T6SS killing across different species, we performed competition assays on solid media. The WT strain exerted a robust inhibitory effect on the growth of *E. coli* DH5α, while the Δ*ypk_0952* mutant faltered in inhibiting the growth of *E. coli* DH5α. Notably, the diminished sensitivity of the Δ*ypk_0952* mutant was substantially reversed through complementation (Fig. 5B; Fig. S7B). In addition, the CFU of DH5α in the WT supernatant was significantly lower than that in the Δ*clpV3* and Δ*ypk_0952* supernatants (Fig. S8). These findings confirm that T6SS-3 exerts inter-species competitive advantage for bacteria in both contact-dependent and contact-independent mechanisms by secreting an antibacterial effector similar to pyocin.

The T6SS has been demonstrated to play a pivotal role in mediating microbial antagonistic interactions, thereby augmenting intestinal colonization (34–36). To determine the ability of YPK_0952 to facilitate *Y. pseudotuberculosis* colonization within the gastrointestinal milieu, mice were subjected to oral infection with uniform inocula

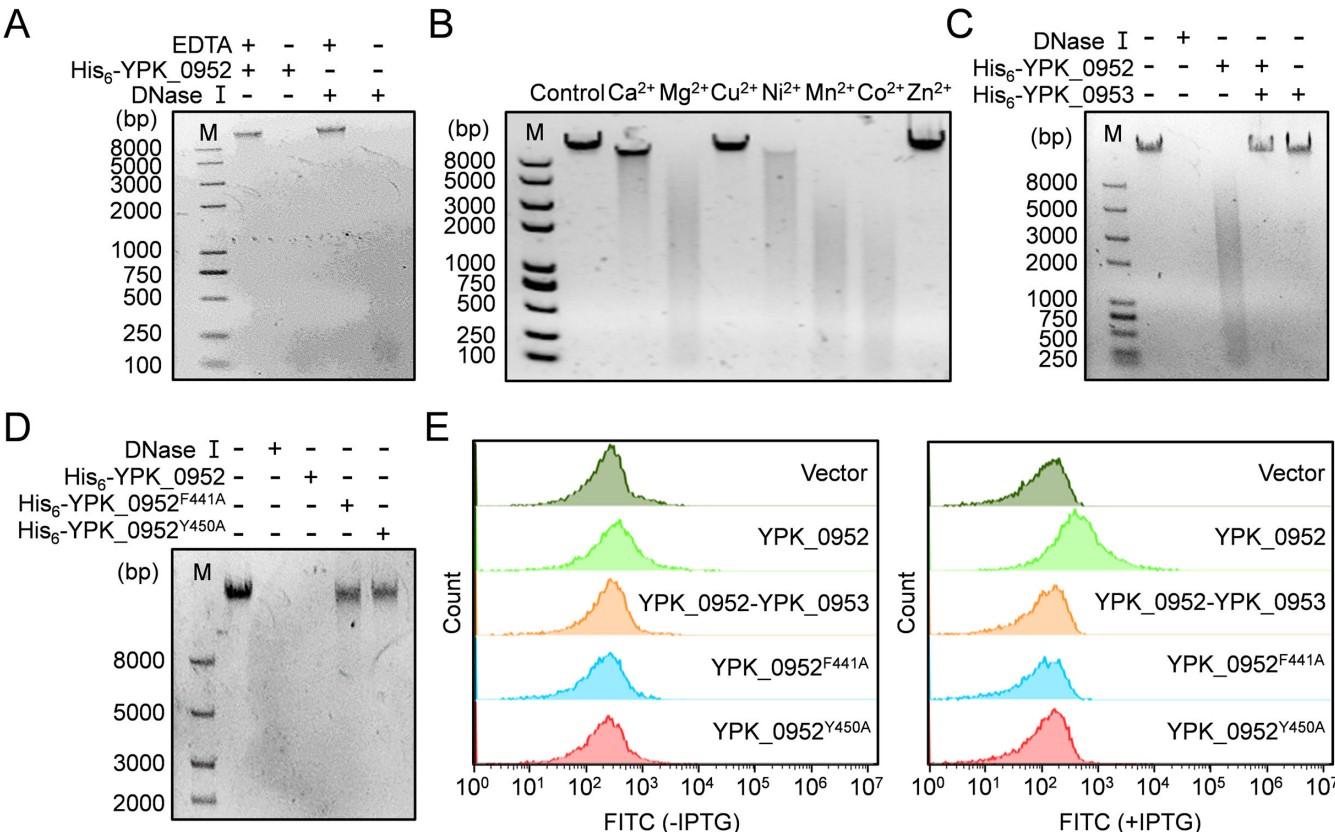

**FIG 4** YPK_0952 exhibits DNase activity. (A) *In vitro* DNase activity assay showed the integrity of λ DNA (0.35 µg) co-incubated with YPK_0952 or DNase I in the reaction buffer with or without EDTA at 37°C for 30 min. Reaction products were analyzed using agarose gel electrophoresis. (B) YPK_0952 and λ DNA co-incubated with the reaction buffer containing different divalent metal ions. (C) YPK_0953 inhibited the DNase activity of YPK_0952. The DNase activity of YPK_0952 and YPK_0953 co-incubated was tested in the reaction buffer. (D) The DNase activity of the YPK_0952$^{F441A}$ and YPK_0952$^{Y450A}$ variants was tested along with YPK_0952 and DNase I in the reaction buffer. (E) YPK_0952-induced genomic DNA fragmentation was detected before (left) and 4 h after (right) IPTG induction in the TUNEL assay. DNA fragmentation was detected based on the monitoring of fluorescence intensity (indicated on the x-axis) using flow cytometry. The counts resulting from cell sorting are shown on the y-axis.

(10$^9$ CFUs) of either the WT *Y. pseudotuberculosis* strain or the Δ*ypk_0952*Δ*ypk_0953* mutant counterpart. The quantification of colonization within the colon and small intestine of the infected murine cohort revealed 24 h post-infection (Fig. 5C). Observations indicated that the counts of the WT strains of *Y. pseudotuberculosis* in the colon and small intestine of infected mice were significantly greater than those of the Δ*ypk_0952*Δ*ypk_0953* strains. These results attest to the indispensability of the YPK_0952-mediated T6SS killing pathway in successful colonization of the mouse gut by *Y. pseudotuberculosis*.

## DISCUSSION

Bacteria have evolved a multitude of mechanisms to compete for resources within complex microbial communities. Control of key nutrients is a central aspect of this survival struggle, for instance, iron is an essential element for cell function, and most bacteria can secrete siderophores to efficiently bind and capture iron ions (37). Beyond nutritional competition, bacteria employ various strategies to optimize their survival capabilities. For example, *Pseudomonas aeruginosa* can utilize its motility to overtake other microorganisms such as *Agrobacterium tumefaciens*, demonstrating how movement can aid bacteria in avoiding or enhancing competitive interactions (38). Chemical warfare is also part of the competitive nature between bacteria, with the production of antibiotics being a common mode of attack (39). However, in the face of antibiotic

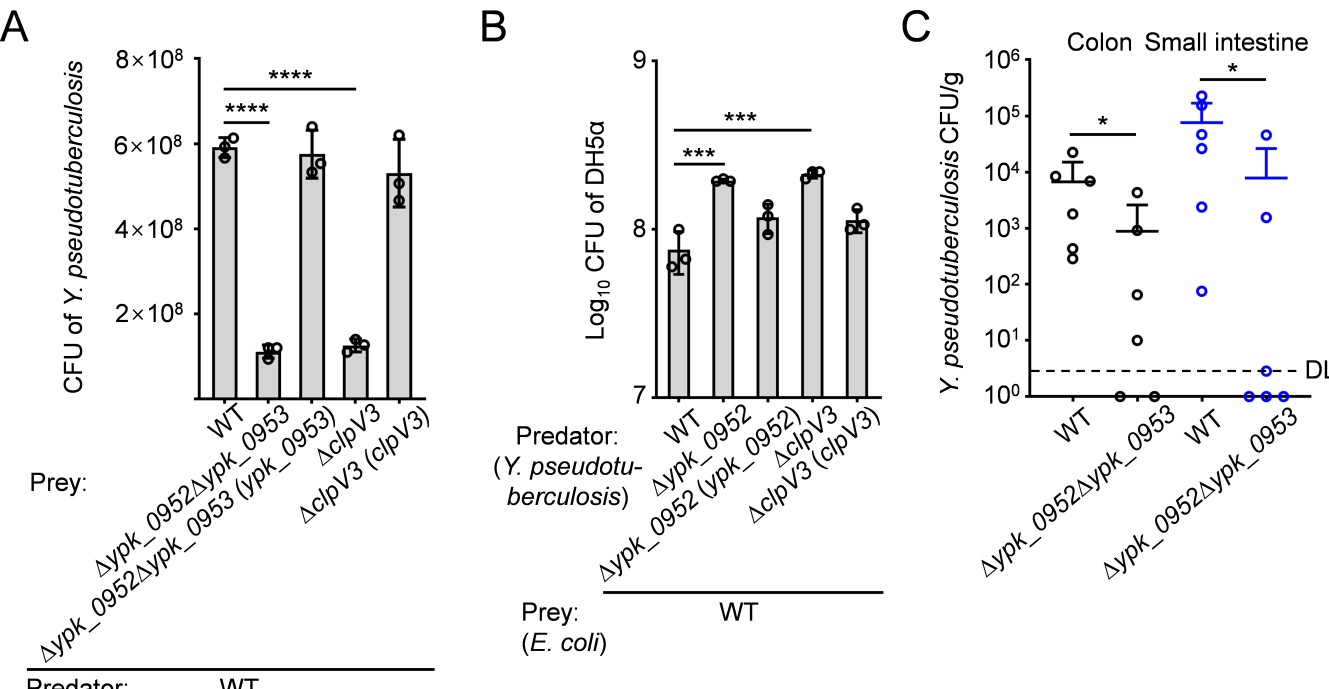

FIG 5 YPK_0952 mediates contact-dependent T6SS killing. (A) Intra-species growth competition between the indicated *Y. pseudotuberculosis* predator and prey strains. Predator and prey strains (1:1) were mixed and then grown for 48 h on a solid support. The CFU of the prey strains was measured based on plate counts. Error bars represent the mean ± SD of prey CFU from three independent experiments, with two-tailed, unpaired Student's *t*-test. *$P < 0.0332$, **$P < 0.0021$, ***$P < 0.0002$, ****$P < 0.0001$. (B) Inter-species growth competition experiments between the indicated *Y. pseudotuberculosis* predator and *E. coli* DH5α prey strains. Predator and prey strains (10:1) were mixed and then grown for 12 h on a solid medium at 26°C. The survival of *E. coli* cells was quantified by counting the CFUs on selective plates. Error bars represent the mean ± SD of three independent experiments, with two-tailed, unpaired Student's *t*-test. *$P < 0.0332$, **$P < 0.0021$, ***$P < 0.0002$, ****$P < 0.0001$. (C) YPK_0952-mediated T6SS killing pathway facilitates *Y. pseudotuberculosis* colonization of mouse gut. Mice ($n = 6$) were orally gavaged with $10^9$ CFUs of indicated *Y. pseudotuberculosis* strains. Animals were sacrificed 24 h after the challenge, and bacterial loads in the colon and small intestine were measured. Error bars represent the mean ± SD of recovered CFUs. Statistical analysis of experiments was carried out using the two-sided Mann–Whitney test. *$P < 0.0332$, **$P < 0.0021$, ***$P < 0.0002$, ****$P < 0.0001$. DL, detection limit.

assault, some bacteria can acquire resistance through genetic mutations and the spread of these genes. Biofilm formation represents another bacterial strategy to counteract antibiotic attacks. The polysaccharide components within biofilms can reduce the penetration of antibiotics, thereby protecting the resident bacteria (40). Additionally, the T6SS is an effective competitive weapon for bacteria that is capable of injecting toxins directly into target cells or secreting it around the target cell before being internalized, and possesses the ability to kill a variety of cell types, including bacterial and eukaryotic cells. In summary, bacteria maintain their niche within microbial communities through a variety of complex survival strategies, constantly adapting to and impacting their environment.

In response to environmental or competitive stress, bacteria often release a repertoire of proteinaceous toxins known as bacteriocins, which are adept at targeting and killing neighboring bacteria (41, 42). These toxins exhibit multifaceted functions, including forming membrane-penetrating pores, acting as nucleases to degrade DNA or RNA within target cells, or disrupting cell wall biosynthesis (33). Most *P. aeruginosa* strains produce pyocins, and each strain can synthesize several types of pyocins (43). The majority of these diverse pyocins possess killing abilities mediated by different mechanisms. For instance, pyocins S1, S2, S3, and AP41 are characterized by DNase activity; tRNase activity is associated with pyocin S4, while channel-forming activity is associated with pyocin S5 (44). Intriguingly, our investigation revealed the presence of S-type pyocin in *Y. pseudotuberculosis*, which is secreted by the T6SS-3 (23). A noteworthy

aspect of bacteriocin involves the self-preservation mechanism in bacteriocin-producing bacteria (32, 45, 46). These bacteria employ an immune protein to shield themselves from the deleterious effects of their bacteriocin, a protective measure that is absent in susceptible strains. Additionally, our results of YPK_0952 revealed demonstrable *in vivo* and *in vitro* DNase activity, which was counteracted by the inhibitory action of the YPK_0953 immune protein.

The investigation of protein bacteriocins has primarily focused on two distinct groups: colicins, which originate from *E. coli* (41), and pyocins, which are produced by *P. aeruginosa* (47). Nucleases of pyocins always have a cytotoxic domain at the carboxyl-terminus that binds homologous immune proteins (43). Through sequence analysis of YPK_0952, we found that the YPK_0952 carboxyl-terminus was the most toxic region. It is known that a variety of metals support DNA cleavage by pyocin DNase (48). Pyocin S8 in *P. aeruginosa* has emerged as an example of a metal-dependent endonuclease; treatment with EDTA completely disrupted its ability to cleave the supercoiled pUC18 plasmid. Furthermore, supplementation of apo S8 with $Ni^{2+}$ robustly induced DNase activity, while $Mn^{2+}$ and $Mg^{2+}$ exhibited more moderate effects, and $Zn^{2+}$ emerged as an inhibitory agent (49). The activity of pyocins in the presence of different metals continues to reveal nuances. Pyocin SX1 exhibited heightened activity in the presence of $Mg^{2+}$ and $Ni^{2+}$, showing its ability to degrade plasmid DNA completely. Conversely, in the presence of $Zn^{2+}$, subduing activity was observed (50). This research unequivocally demonstrated that YPK_0952 is an active metal-dependent nuclease that targets DNA. Metal-dependent cleavage assays confirmed that YPK_0952 can cleave double-stranded DNA with a single metal ion cofactor, including $Mg^{2+}$, $Ni^{2+}$, $Mn^{2+}$, and $Co^{2+}$. This finding underscores the ability of YPK_0952 to leverage diverse divalent metal ions to exert DNase activity.

Increasing evidence indicates the predominance of competitive interactions among bacteria in many environments (51). Gram-negative bacteria employ the T6SS as a contact-dependent antagonistic mechanism, mediating the intoxication of other bacteria or eukaryotic cells (52). This intricate molecular apparatus secretes antibacterial effector proteins by undergoing cycles of extension and contraction (53), which facilitate the targeted delivery of toxic effector proteins into neighboring cells (32, 54). Generally, pyocins exhibit strain-specific bactericidal activity against *P. aeruginosa*, selectively killing other strains within the same species; these strains tend to have a narrow killing spectrum (55). The T6SS is a pivotal determinant of bacterial survival upon contact between two Gram-negative bacterial cells (32). In this study, we found that YPK_0952 showed competitive intra-species and inter-species advantages through T6SS. In contrast to the specificity of pyocin sterilization in *P. aeruginosa*, YPK_0952 can be modified by T6SS-mediated cross-species contact sterilization. YPK_0952 can achieve intra-species and inter-species bactericidal effects through both contact-dependent and contact-independent mechanisms mediated by the T6SS. Furthermore, bacteroides employ the T6SS to colonize and persist in the colon competitively (56). Here, we report a T6SS pyocin-like effector, YPK_0952, which significantly enhances the colonization of *Y. pseudotuberculosis* in the colon and small intestine.

S-type pyocin loci typically encode two protein components: a high-molecular-weight effector protein and a low-molecular-weight immunity protein (57). The effector protein usually consists of three to four domains, with the N-terminus domain involved in recognition of cell surface receptors, the function of the second domain currently unclear, the third domain responsible for pyocin translocation and penetration, and the C-terminus domain possessing lethal activity (43, 44). Like most bacteriocins, S-type pyocins bind to specific receptors on the outer membrane of closely related bacterial species, then translocate across the membrane, and kill their targets. Studies indicate that most S-type pyocins bind to TonB-dependent outer membrane receptors on bacterial cells. For example, pyocins S2 and S4 enter cells through the FpvAI protein (57), pyocin S3 targets FpvAII (58), while pyocin S5 recognizes the FptA ferripyochelin receptor (59). This study found that *Y. pseudotuberculosis* T6SS-3 gains

a contact-independent competitive advantage by secreting the pyocin-like effector, YPK_0952, into the extracellular space, with the T6SS mechanism acting as a conduit for delivering effectors to the outer membrane receptors of prey cells. However, the internalization mechanism of YPK_0952 remains unclear, whether it resembles the cell membrane receptor-mediated mechanism of pyocins, will be further investigated in future studies.

Bacteriocins are antibacterial proteins that selectively kill the phylogenetic kin of their producers. The latent promise of bacteriocins is that they are potential remedies for combating multidrug-resistant bacterial infections (60, 61). However, an imperative prerequisite for harnessing bacteriocins as efficacious antimicrobial agents is to understand how they are imported. Moreover, these important pathways might reveal further processes that could be exploited for newly designed drugs or chimeric bacteriocins endowed with augmented toxicity (62). As protein bacteriocins have begun to be exploited as therapeutics for multidrug-resistant bacteria, their import mechanisms need to be investigated (61). In this study, the pyocin-like effector secreted by the T6SS in *Y. pseudotuberculosis* was identified and characterized as an effector that exerts toxicity through the T6SS. This novelty is crucial for developing an efficacious therapeutic approach that is an alternative to antibiotics.

## ACKNOWLEDGMENTS

We thank the Life Science Research Core Services, Northwest A&F University (Ningjuan Fan, Xiyan Chen, Hui Duan, and Lanlan Wei) for technical support.

This work was supported by the grant of the National Key R&D Program of China (2022YFD1400200 to Y.W.) and the National Natural Science Foundation of China (31970114 and 32170130 to Y.W., and 31670053 and 31725003 to X.S.).

Y.W. and L.Y. conceived the project. L.Y., S.J., S.S., L.W., B.Z., M.Z., Y.Y., and M.Y. performed the experimental work. L.Y., M.Y., X.S., J.P., and Y.W. wrote the manuscript. A.M.F reviewed and edited the manuscript. Y.W. supervised the study. All authors discussed the results and commented on the manuscript.

## AUTHOR AFFILIATIONS

[1]State Key Laboratory for Crop Stress Resistance and High-Efficiency Production, Shaanxi Key Laboratory of Agricultural and Environmental Microbiology, College of Life Sciences, Northwest A&F University, Yangling, Shaanxi, China
[2]College of Life Sciences, Tarim University, Alar, Xinjiang, China
[3]State Key Laboratory for Crop Stress Resistance and High-Efficiency Production, College of Plant Protection, Northwest A&F University, Yangling, Shaanxi, China
[4]Department of Plant Science and Crop Protection, University of Nairobi, Nairobi, Kenya

## AUTHOR ORCIDs

Leilei Yang  http://orcid.org/0009-0006-6389-6125
Xihui Shen  http://orcid.org/0000-0001-6867-8887
Junfeng Pan  http://orcid.org/0000-0001-8666-4446
Yao Wang  http://orcid.org/0000-0002-7149-4234

## FUNDING

| Funder | Grant(s) | Author(s) |
|---|---|---|
| The National Key R&D Program of China | 2022YFD1400200 | Yao Wang |
| The National Natural Science Foundation of China | 31970114, 32170130, 31670053, 31725003 | Xihui Shen<br>Yao Wang |

## AUTHOR CONTRIBUTIONS

Leilei Yang, Conceptualization, Data curation, Formal analysis, Investigation, Methodology, Software, Validation, Visualization, Writing – original draft | Shuangkai Jia, Investigation, Methodology | Sihuai Sun, Investigation, Methodology | Lei Wang, Investigation, Methodology | Bobo Zhao, Investigation, Methodology | Mengsi Zhang, Investigation, Methodology | Yanling Yin, Investigation, Methodology | Mingming Yang, Investigation, Writing – original draft | Alex M. Fulano, Writing – review and editing | Xihui Shen, Writing – original draft | Junfeng Pan, Writing – original draft | Yao Wang, Conceptualization, Data curation, Formal analysis, Funding acquisition, Project administration, Resources, Supervision, Writing – original draft

## DATA AVAILABILITY

All relevant data supporting this study's findings are included in the article and its Supplemental files or from the corresponding authors upon request.

## ETHICS APPROVAL

All mouse experiments were carried out in compliance with the State Council of the People's Republic of China's Regulations for the Administration of Affairs Concerning Experimental Animals. Northwest A&F University's Animal Welfare and Research Ethics Committee authorized the protocol (protocol number NWAFUSM2018001). Six-week-old female mice (BALB/c) were acquired from Xi'an Jiaotong University's central animal laboratory (Xi'an, China) and were kept at a temperature of $24 \pm 2°C$, $50 \pm 10\%$ humidity, 35 exchange air flow, in a light-controlled room (12-h light, 12-h darkness) with unrestricted access to food and water.

## ADDITIONAL FILES

The following material is available online.

### Supplemental Material

**Supplemental material (Spectrum04278-23-s0001.docx).** Fig. S1 to S8; Table S1 and S2.

### Open Peer Review

**PEER REVIEW HISTORY (review-history.pdf).** An accounting of the reviewer comments and feedback.

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
