## [Reviewer comments · Microbiology Spectrum]

Microbiology Spectrum

A pyocin-like T6SS effector mediates bacterial competition in *Yersinia pseudotuberculosis*

Leilei Yang, Shuangkai Jia, Sihuai Sun, Lei Wang, Bobo Zhao, Mengsi Zhang, Yanling Yin, Mingming Yang, Alex M. Fulano, Xihui Shen, Junfeng Pan, and Yao Wang

Corresponding Author(s): Yao Wang, Northwest A&F University College of Life Sciences

Review Timeline:

Submission Date:	December 22, 2023
Editorial Decision:	February 5, 2024
Revision Received:	March 14, 2024
Editorial Decision:	April 1, 2024
Revision Received:	April 16, 2024
Accepted:	April 18, 2024

Editor: Jing Han

Reviewer(s): The reviewers have opted to remain anonymous.

Transaction Report:

DOI: <https://doi.org/10.1128/spectrum.04278-23>

Re: Spectrum04278-23 (A pyocin-like T6SS effector mediates bacterial competition in *Yersinia pseudotuberculosis*)

Dear Prof. Yao Wang:

Thank you for the privilege of reviewing your work. Below you will find my comments, instructions from the Spectrum editorial office, and the reviewer comments.

Revision Guidelines

Sincerely,
Jing Han
Editor
Microbiology Spectrum

Reviewer #1 (Comments for the Author):

Yang and coworkers aim to shed light into the mode of action of a putative T6SS effector encoded within one of the T6SS clusters of *Y. pseudotuberculosis*. They leveraged molecular microbiology and biochemistry methods to establish the mode of action. In vivo work challenging a gut infection model showed the contribution of this effector to virulence.

General comment.

There is a gap in knowledge in the characterization of *Y. pseudotuberculosis* T6SS effectors. This analysis may unveil new principles of general interest beyond *Y. pseudotuberculosis*. The presented work is sounded although the authors do not provide conclusive evidence demonstrating that the effector is indeed secreted in a T6SS-dependent manner, and that the effector exerts a killing effect when delivered via the T6SS in a contact-dependent manner. These two experiments are crucial to sustain their conclusions. In addition some other controls and clarifications are needed. This reviewer hopes the authors will find the comments useful to strengthen this manuscript.

Major comments.

1. No static analysis is done in the case of figures 2 and 4.
2. Authors have not demonstrated the secretion of the effector in a T6SS-3 dependent manner. This experiment is easily done using a tagged version of the protease (either in the chromosome or in a plasmid) and assess the secretion to the supernatant by the wild-type and the T6SS-3 null (non-functional) mutant strain. In case, there is still secretion (or lower), authors should then assess the secretion by any of the other T6SS systems.
3. A point the authors make is the contact-dependent delivery of the toxin. This is important for this type of pyocin-like toxins that can exert their antimicrobial effect without need of being injected. As such their results could be interpreted by T6SS secretion of the toxin, and its subsequent internalization from the supernatant instead of being injected into the cell. Some work should be done to provide evidence. This reviewer suggests these two experiments: (i) separate prey and predator by a membrane and assess killing; (ii) test the antimicrobial properties (and capacity to degrade DNA) of condition media from wild-type, T6SS-3 null strain, and the effector mutant. Authors may wish to consider assessing the contact-dependent translocation by microscopy.
4. Pull-down experiments. Could the authors explain the two bands observed in the anti-his blot?
5. Fig 4. Panel A does not show any effect for the T6SS effector. Whereas the results with EDTA may support the role of divalent cations, panel B cannot be considered conclusive for the role of the specific cations (the buffer may be "contaminated"). Besides there are differences in the outcome of the different cations, and, for example, it might be tempting to speculate that this is a Mn toxin. Further robust biochemistry experiments are warranted to clarify this point. Panel E; the signal of the vector alone (as well as other controls) is unusually high for a flow cytometry experiment. Authors need to provide as supplementary material the gating showing the particle size and the FITC channel. In addition, other technique is needed to substantiate these observations (for example immunoblotting).
6. Fig 5A is bit confusing. On the one hand, this reviewer suggests to show the counts of ONLY the prey. In this context, wt versus wt there should not be any differences as this would be the max value. Potentially, this result should be similar to WT-clpV3. If here you have more bacteria then authors need to speculate/explain why this is the case because these strains should contain the immunity proteins. In panels A and B, it will be informative to show the control of prey incubated for the same time with no other bacteria but the same volume of PBS.
6. Authors do not provide any rationale for the chosen site-directed mutants generated. Are these residues within the active site? any other reason? This should be specified in the text, and if needed provide some modeling information.
7. Authors need to revise extensively the manuscript to increase the readability by avoiding superfluous expressions, avoid long sentences, limit the use of passive voice, and ensure there are no grammatical errors. There are too many to pinpoint but just as examples: lines 65-67, lines 71-75, line 77-79, lines 228-230, line 370, line 414 and many others.

Minor comments.

1. Authors should consider including as supplementary material a figure showing the T6SS-3 locus and the positions of the genes under consideration in this study.
2. Fig 5. Donor and recipient are not the most used terminology for these assays. Please use instead prey and predator.
3. In vivo work. Authors need to show in the graph the limit of the detection of the plating.
4. Authors need to justify why they had only infected female. Infecting only one sex is not a good practice.

Reviewer #2 (Comments for the Author):

1. The overall structure and logic of the article are clear, but the content is limited. I hope the author can further explore the mechanism of mediating bacterial competition.
2. The writing of this manuscript is rough, for example, the description of certain experimental methods is too simple and not suitable for researchers to conduct experimental operations according to the current methods.
3. As for mouse colonization determination, I suggest supplementing experimental conditions such as the number of mice in each group, the weight of colon and small intestine tissues, etc.

如377，至于小鼠定植测定，我建议补充实验条件，如每组小鼠数量、结肠和小肠组织的重量等。

Response to reviewers

We are pleased to see the reviewers' enthusiasm toward our manuscript and appreciate their insightful suggestions that have helped strengthen the revised manuscript. Here, we provide a detailed response to each reviewer's comments and the corresponding changes made to the revised manuscript. Please note the following: Text in black font corresponds to the reviewers' comments, blue font text is our response to the reviewers' comments, and highlighted text represents the excerpts from the revised manuscript.

Response to Reviewer 1

Yang and coworkers aim to shed light into the mode of action of a putative T6SS effector encoded within one of the T6SS clusters of *Y. pseudotuberculosis*. They leveraged molecular microbiology and biochemistry methods to establish the mode of action. In vivo work challenging a gut infection model showed the contribution of this effector to virulence.

Re: We greatly appreciate the reviewer's invaluable comments and suggestions on our work, which are extremely helpful for us to improve our manuscript.

General comment

There is a gap in knowledge in the characterization of *Y. pseudotuberculosis* T6SS effectors. This analysis may unveil new principles of general interest beyond *Y. pseudotuberculosis*. The presented work is sounded although the authors do not provide conclusive evidence demonstrating that the effector is indeed secreted in a T6SS-dependent manner, and that the effector exerts a killing effect when delivered via the T6SS in a contact-dependent manner. These two experiments are crucial to sustain their conclusions. In addition some other controls and clarifications are needed. This reviewer hopes the authors will find the comments useful to strengthen

this manuscript.

Re: Thanks for the positive comments and constructive suggestions! The dependence of YPK_0952 on T6SS-3 for secretion and its killing effect via T6SS in a contact-dependent manner are very important to our conclusion.

- 1) The reliance of YPK_0952 on T6SS-3 secretion has been confirmed by our group and published in the reference 23 of the manuscript, where the corresponding description in the manuscript, page 14, " The primary open reading frame (ORF), YPK_0952, is distinguished by a typical PAAR domain at its N-terminus and an S-type pyocin domain at its C-terminus. YPK_0952 predominantly functions as a T6SS effector secreted by T6SS-3 (23).", and the literature describes it as " To identify novel T6SS effectors, we searched the *Yptb* YPIII genome for genes containing the Proline-Alanine-Alanine-Rginine (PAAR) domain, a conserved effector-targeting domain that is linked or adjacent to numerous known T6SS effectors^{7,21,22}. A gene locus encoding multiple hypothetical T6SS effector-immunity pairs was identified (YPK_0952-0958, Fig. 1a). Both the first and last open-reading frame (ORF) of this locus contain PAAR domains. The first ORF, YPK_0952, contains a typical PAAR domain at its N-terminus and an S-type pyocin domain at its C-terminus. When VSVG-tagged YPK_0952 was produced in YPIII, the secreted protein was readily detected in the supernatant. However, YPK_0952 secretion was abrogated in the $\Delta 4cplV$ mutant, in which all four essential ATPase genes in the four sets of T6SSs were deleted, strongly suggesting that YPK_0952 is a T6SS effector. The secretion of YPK_0952 was dramatically diminished with deletion of *cplV3*, but not with deletion of *cplV1*, *cplV2*, or *cplV4*, further indicating that YPK_0952 is a T6SS effector mainly associated with T6SS-3 (Supplementary Fig. 1a). Similarly, we showed that YPK_0954, which does not contain a PAAR domain but is located downstream of YPK_0952, is also a T6SS-3 effector (Fig. 1b and Supplementary Fig. 1b)."
- 2) The intra-species contact competition experiment (Fig. 5A) shows that the wild-type *Y. pseudotuberculosis* has a distinct advantage in competition with $\Delta ypk_0952\Delta ypk_0953$ (YPK_0952 and YPK_0953 deletion mutant) and $\Delta cplV3$

(*clpV3* deletion mutant). The inter-species contact competition experiment (Fig. 5B) demonstrates that both Δypk_0952 and $\Delta clpV3$ have a significantly lower competitive advantage over *E. coli* DH5 α compared to the wild-type *Y. pseudotuberculosis*. It is evident from this that YPK_0952 exerts its lethal effect by T6SS in a contact-dependent manner.

Comment 1: No statically analysis is done in the case of figures 2 and 4.

Re: Thank you for raising this important issue. For the experiments involved in Fig. 2 and Fig. 4, we referred to the methods described in reference 23 from the manuscript.

- 1) Fig. 2 presents the toxicity growth experiments of *Y. pseudotuberculosis* and *E. coli*, where the number of bacteria increases to varying degrees as the culture time extends. We conducted statistical analyses, the data in the manuscript represent mean \pm SD, $n = 3$, with three biological replicates.
- 2) Fig. 4E showcases a flow cytometry assay, where each sample was tested for ten thousand cells, meeting the requirements for a TUNEL assay. The results obtained are accurate and reliable.

Comment 2: Authors have not demonstrated the secretion of the effector in a T6SS-3 dependent manner. This experiment is easily done using a tagged version of the protean (either in the chromosome e or in a plasmid) and assess the secretion to the supernatant by the wild-type and the T6SS-3 null (non-functional) mutant strain. In case, there is still secretion (or lower), authors should then assess the secretion by any of the other T6SS systems.

Re: Thanks for the kind reminder from the reviewer. Our group's earlier work has already demonstrated that YPK_0952 is secreted by T6SS-3, as mentioned on page 14 of the manuscript. The researchers examined the secretion of YPK_0952 in the wild-type and various T6SS mutants. It was discovered that the secretion of YPK_0952 was completely undetectable in mutants lacking all four sets of T6SS, and significantly reduced in the T6SS-3 mutant, indicating that YPK_0952 is secreted via T6SS-3. The description in manuscript reference 23 is as follows:

“Supplementary Figure 1: YPK_0954 (Tce1) is a T6SS-3 secreted nuclease effector and YPK_0955 (Tci1) is its immunity protein. a-b, YPK_0952 and YPK_0954 are T6SS-3 effectors. Plasmids directing the expression of YPK_0952-VSVG (a) or YPK_0954-VSVG (b) were introduced into the indicated *Y. pseudotuberculosis* strains. Total cell pellet (Pellet) and secreted proteins in culture supernatant (Sup) were isolated and probed for the presence of the fusion protein. The cytosolic RNA polymerase (RNAP) or isocitrate dehydrogenase (ICDH) was similarly detected as a control.”

Comment 3: A point the authors make is the contact-dependent delivery of the toxin. This is important for this type of pyocin-like toxins that can exert their antimicrobial effect without need of being injected. As such their results could be interpreted by t6SS secretion of the toxin, and its subsequent internalization from the supernatant instead of being injected into the cell. Some work should be done to provide evidence. This reviewer suggests these two experiments: (i) separate prey and predator by a membrane and assess killing; (ii) test the antimicrobial properties (and capacity to degrade DNA) of condition media from wild-type, t6SS-3 null strain, and the effector mutant. Authors may wish to consider assessing the contact-dependent translocation by microscopy.

Re: We thank the reviewer for the supportive comments on our work. The manuscript

reports that the pyocin-like effector protein YPK_0952 is delivered into other cells by T6SS through contact-dependent delivery, exerting its bactericidal effect. In contrast, contact-independent delivery does not require injection and completes toxin transfer through cell surface receptors. Therefore, the experiment involving the killing of bacteria by internalization of supernatant containing the YPK_0952 toxin belongs to contact-independent delivery of toxins, and is not part of the contact-dependent competition mentioned in our text.

Comment 4: Pull-down experiments. Could the authors explain the two bands observed in the anti-his blot?

Re: Thank you very much for your insightful comments and suggestions to improve our pull-down results! The observation of two bands in the anti-His blot may be due to impurities in the His₆-YPK_0952 protein used in the GST pull-down experiments. We have repurified the His₆-YPK_0952 protein and conducted the GST pull-down experiment again. The revised results have been updated in the manuscript on page 27. The updated Fig. 3A is as follows.

Figure 3. Direct binding between YPK_0952 and YPK_0953 was detected using GST pull-down and bacterial two-hybrid assays.

(A) His₆-YPK_0952 was incubated with GST-YPK_0953, GST, or an irrelevant recombinant protein GST-YPK_3549, and the protein complexes captured on glutathione beads were detected using western blotting. (B) Interactions were assessed using MacConkey maltose plates (upper) and the β -galactosidase assay

(lower). Error bars represent the mean \pm SD of three independent experiments, with two-tailed, unpaired Student's *t*-test. **P* < 0.0332; ***P* < 0.0021; ****P* < 0.0002, *****P* < 0.0001.

Comment 5: Fig 4. Panel A does not show any effect for the t6SS effector. Whereas the results with EDTA may support the role of divalent cations, panel B cannot be considered conclusive for the role of the specific cations (the buffer may be "contaminated"). Besides there are differences in the outcome of the different cations, and, for example, it might be tempting to speculate that this is a Mn toxin. Further robust biochemistry experiments are warranted to clarify this point. Panel E; the signal of the vector alone (as well as other controls) is unusually high for a flow cytometry experiment. Authors need to provide as supplementary material the gating showing the particle size and the FITC channel. In addition, other technique is needed to substantiate these observations (for example immunoblotting).

Re: We appreciate the valuable suggestions.

1) As shown in Fig. 4A, His₆-YPK_0952 is capable of degrading DNA, and the experimental results are the same as those with the addition of DNase I alone, thereby proving that His₆-YPK_0952 is a DNase. As for your concerns regarding Fig. 4B, we have reformulated the reaction buffer (20 mM MES, 100 mM NaCl, pH 6.9) and conducted the experiment with only the corresponding metal ions added to each lane, ensuring that the buffer was not contaminated. At the same time, we set up a control by adding only λ DNA to the lane to demonstrate that the reaction buffer was not contaminated. Furthermore, our intention is to demonstrate that YPK_0952 is a DNase that relies on various divalent metal cations, rather than a specific type of divalent metal cation enzyme (such as, Mn²⁺). The updated Fig. 4B on page 28 of the manuscript is as follows.

Figure 4. YPK_0952 exhibits DNase activity.

(A) *In vitro* DNase activity assay showed the integrity of λ DNA (0.35 μ g) co-incubated with YPK_0952 or DNase I in the reaction buffer with or without EDTA at 37°C for 30 min. Reaction products were analyzed using agarose gel electrophoresis. (B) YPK_0952 and λ DNA co-incubated with the reaction buffer containing different divalent metal ions. (C) YPK_0953 inhibited DNase activity of YPK_0952. The DNase activity of the YPK_0952 and YPK_0953 co-incubated were tested in the reaction buffer. (D) The DNase activity of the YPK_0952^{F441A} and YPK_0952^{Y450A} variants was tested along with YPK_0952 and DNase I in the reaction buffer. (E) YPK_0952-induced genomic DNA fragmentation was detected before (Left) and 4 h after (Right) IPTG induction in the TUNEL assay. DNA fragmentation was detected based on the monitoring of fluorescence intensity (indicated on the x-axis) using flow cytometry. The counts resulting from cell sorting are shown on the y-axis.

2) We have added gating for the FITC channel and display granularity as supplementary material in the flow cytometry experiment, found in the supplementary information on page 5. The updated Supplementary Figure 4 in the supplementary information on page 5 is as follows.

Supplementary Figure 4: Flow cytometry experiment FITC gating diagrams. In each figure, the bacterial morphology and size are similar, with the same gating strategy applied (-IPTG, 0 h; +IPTG, induced by IPTG for 4 h).

3) *E. coli* BL21 (DE3) containing plasmids with pET28a, pET28a-*ypk_0952*, pET28a-*ypk_0952-ypk_0953*, pET28a-*ypk_0952*^{F441A} or pET28a-*ypk_0952*^{Y450A} were induced with 0.5 mM IPTG for 4 hours followed by western blotting. The results showed that all toxin were expressed except for the empty vector. Fig. 4E in the manuscript demonstrates that only the YPK_0952 was TUNEL positive. This indicates that only YPK_0952 has DNase activity that can fragment DNA within *E. coli* BL21 (DE3).

Determination of YPK_0952 expression levels in *E. coli* BL21 (DE3) harboring pET28a or its derivatives using an anti-His antibody.

Comment 6: Fig 5A is bit confusing. On the one hand, this reviewer suggests to show the counts of ONLY the prey. In this context, wt versus wt there should not be any differences as this would be the max value. Potentially, this result should be similar to WT-clpV3. If here you have more bacteria then authors need to speculate/explain why this is the case because these strains should contain the immunity proteins. In panels

A and B, it will be informative to show the control of prey incubated for the same time with no other bacteria but the same volume of PBS.

Re: Sorry for the imprecise description in the former manuscript.

- 1) We have made changes to Figure 5A, now only displaying the CFU of prey, on page 29.
- 2) Regarding the question of why the WT has more bacteria after competing with the $\Delta clpV3$ mutant that contains the immune protein YPK_0953. We think that in T6SS-mediated contact-dependent competition, the $\Delta clpV3$ mutant's inability to secrete effector proteins into predator strains prevents it from counterattacking. Therefore, although the prey $\Delta clpV3$ mutant contains the immune protein YPK_0953, the predator (WT) can still attack the prey ($\Delta clpV3$) through contact-dependent competition.

Figure 5. YPK_0952 mediates contact-dependent T6SS killing.

(A) Intra-species growth competition between the indicated *Y. pseudotuberculosis* predator and prey strains. Predator and prey strains (1:1) were mixed and then grown for 48 h on a solid support. The CFU of the prey strains was measured based on plate counts. Error bars represent the mean ± SD of prey CFU from three independent experiments, with two-tailed, unpaired Student's *t*-test. **P* < 0.0332; ***P* < 0.0021; ****P* < 0.0002, *****P* < 0.0001. (B) Inter-species growth competition experiments between the indicated *Y. pseudotuberculosis* predator and *E. coli* DH5α prey strains.

Predator and prey strains (10:1) were mixed and then grown for 12 h on a solid medium at 26 °C. The survival of *E. coli* cells was quantified by counting CFUs on selective plates. Error bars represent the mean \pm SD of three independent experiments, with two-tailed, unpaired Student's *t*-test. **P* < 0.0332; ***P* < 0.0021; ****P* < 0.0002, *****P* < 0.0001. (C) YPK_0952-mediated T6SS killing pathway facilitates *Y. pseudotuberculosis* colonization of mouse gut. Mice (*n* = 6) were orally gavaged with 10⁹ CFUs of indicated *Y. pseudotuberculosis* strains. Animals were sacrificed 24 h after the challenge, and bacterial loads in the colon and small intestine were measured. Error bars represent the mean \pm SD of recovered CFUs. Statistical analysis of experiments were carried out using the two-sided Mann-Whitney test. **P* < 0.0332; ***P* < 0.0021; ****P* < 0.0002, *****P* < 0.0001. DL, detection limit.

Comment 7: Authors do not provide any rationale for the chosen site-directed mutants generated. Are these residues within the active site? any other reason? This should be specified in the text, and if needed provide some modeling information.

Re: Thank you for your suggestion! The manuscript mentions two site-directed mutants, YPK_0952^{F441A} and YPK_0952^{Y450A}, which are located within the toxic activity region. Fig. 1A demonstrates that F441 and Y450 are highly conserved sites in YPK_0952, and Fig. 2C and 2D show that the amino acid region from position 411 to 510 is its toxic region. By performing site-directed mutagenesis toxicity screening on multiple highly conserved sites within this toxic region, we found that mutations at F441 and Y450 result in the loss of YPK_0952 toxicity, thereby identifying these two sites as key toxic amino acids. We have added the site-directed mutagenesis toxicity screening experiment to the updated supplementary information, Supplementary Figure 2, on page 3.

Supplementary Figure 2: Growth curves of *E. coli* BL21 (DE3) harboring indicated plasmids. The growth of the indicated strains in LB was monitored by measuring OD₆₀₀ at indicated time points. Error bars represent the mean ± standard deviation (SD) of three independent experiments.

Comment 8: Authors need to revise extensively the manuscript to increase the readability by avoiding superfluous expressions, avoid long sentences, limit the use of passive voice, and ensure there are no grammatical errors. There are too many to pinpoint but just as examples: lines 65-67, lines 71-75, line 77-79, lines 228-230, line 370, line 414 and many others.

Re: Thanks for the kind reminder from the reviewer. We made a detailed revision of the manuscript.

Minor comments:

1. Authors should consider including as supplementary material a figure showing the T6SS-3 locus and the positions of the genes under consideration in this study.

Re: Thank you very much for your insightful comments. We have added the gene atlas of the *Y. pseudotuberculosis* T6SS to the updated supplementary information, on page 2.

Y. pseudotuberculosis T6SS gene cluster

Supplementary Figure 1: Schematic of a gene cluster encoding T6SS effectors in *Y. pseudotuberculosis*. The arrows indicate the position and transcription direction of each gene on the T6SS gene cluster. Locus tag numbers are provided on the top of each gene, and the genes *ypk_0952* and *ypk_0953* are represented by green and yellow, respectively.

2. Fig 5. Donor and recipient are not the most used terminology for these assays. Please use instead prey and predator.

Re: We appreciate the valuable suggestion. The term “donor” and “recipient” has been changed into “predator” and “prey” throughout the revised manuscript and supporting information. Fig. 5 has been updated in the manuscript, on page 29.

3. In vivo work. Authors need to show in the graph the limit of the detection of the plating.

Re: Thank you very much for your helpful comments. We have updated Fig. 5C in the

4. Authors need to justify why they had only infected female. Infecting only one sex is not a good practice.

Re: Thank you very much for your positive comments. Bacterial gut colonization experiments are not related to the sex of the experimental animals. In the studies by Mougous, and Song et al. (Mougous* et al. *Nature* 2019, 575, 224-228; Song et al. *Nat. Commun.* 2021, 12, 423-434.), female mice were used for gut bacterial colonization experiments. Therefore, our choice of docile and easy-to-handle female mice for our experiments does not affect the conclusions of the experiment.

Response to Reviewer 2

Comment 1: The overall structure and logic of the article are clear, but the content is limited. I hope the author can further explore the mechanism of mediating bacterial competition.

Re: We appreciate the kind reminder! We further explored the mechanisms of mediating bacterial competition and have included corresponding descriptions in the updated manuscript, on page 17: "Bacteria have evolved a multitude of mechanisms

to compete for resources within complex microbial communities. Control of key nutrients is a central aspect of this survival struggle, for instance, iron is an essential element for cell function, and most bacteria can secrete siderophores to efficiently bind and capture iron ions (37). Beyond nutritional competition, bacteria employ various strategies to optimize their survival capabilities. For example, *P. aeruginosa* can utilize its motility to overtake other microorganisms such as *Agrobacterium tumefaciens*, demonstrating how movement can aid bacteria in avoiding or enhancing competitive interactions (38). Chemical warfare is also part of the competitive nature between bacteria, with the production of antibiotics being a common mode of attack (39). However, in the face of antibiotic assault, some bacteria can acquire resistance through genetic mutations and the spread of these genes. Biofilm formation represents another bacterial strategy to counteract antibiotic attacks. The polysaccharide components within biofilms can reduce the penetration of antibiotics, thereby protecting the resident bacteria (40). Additionally, the T6SS is an effective competitive weapon for bacteria that is capable of injecting toxins directly into target cells and possesses the ability to kill a variety of cell types, including bacterial and eukaryotic cells. In summary, bacteria maintain their niche within microbial communities through a variety of complex survival strategies, constantly adapting to and impacting their environment.”

Comment 2: The writing of this manuscript is rough, for example, the description of certain experimental methods is too simple and not suitable for researchers to conduct experimental operations according to the current methods.

Re: Thanks for the kind reminder from the reviewer. We have revised the experimental methods in the manuscript, providing detailed descriptions to ensure researchers can replicate the experiments using the current methods.

Comment 3: As for mouse colonization determination, I suggest supplementing experimental conditions such as the number of mice in each group, the weight of colon and small intestine tissues, etc.

Re: We appreciate the valuable suggestion. In the updated manuscript, we have added a description related to the “Murine colonization assay”, on page 13: “Female 6-week-old BALB/c mice were acclimated in the laboratory for 3 days. Mice were orally gavaged with 10^9 CFUs of the indicated *Y. pseudotuberculosis* strains and monitored for 24 h. At the end of the experiment, six mice from each group were euthanized, and colon tissues (from the WT group: 0.43 g, 0.29 g, 0.39 g, 0.34 g, 0.35 g, 0.36 g; from the $\Delta ypk_{0952}\Delta ypk_{0953}$ group: 0.32 g, 0.51 g, 0.39 g, 0.33 g, 0.33 g, 0.36 g) and small intestine tissues (from the WT group: 0.95 g, 1.07 g, 0.85 g, 0.9 g, 0.81 g, 0.76 g; from the $\Delta ypk_{0952}\Delta ypk_{0953}$ group: 0.84 g, 0.96 g, 1.14 g, 0.86 g, 1.49 g, 1.1 g) were weighed. The colon and small intestine tissues were then homogenized in 0.5 mL of PBS on ice, and samples of different dilutions were plated on selective YLB agar containing appropriate antibiotics for CFU counting (23). This was followed by subsequent absolute quantification of CFU by normalization of each sample to the initial pellet weight.”

References

1. Ross Benjamin D, Verster Adrian J, Radey Matthew C, Schmidtke Danica T, Pope Christopher E, Hoffman Lucas R, Hajjar Adeline M, Peterson S Brook, Borenstein Elhanan, Mougous Joseph D. 2019. Human gut bacteria contain acquired interbacterial defence systems. *Nature*. 575:224-228.
2. Song L, Pan J, Yang Y, Zhang Z, Cui R, Jia S, Wang Z, Yang C, Xu L, Dong TG, Wang Y, Shen X. 2021. Contact-independent killing mediated by a T6SS effector with intrinsic cell-entry properties. *Nat. Commun*. 12:423-434.

Re: Spectrum04278-23R1 (A pyocin-like T6SS effector mediates bacterial competition in *Yersinia pseudotuberculosis*)

Dear Prof. Yao Wang:

Thank you for the privilege of reviewing your work. Below you will find my comments, instructions from the Spectrum editorial office, and the reviewer comments.

Revision Guidelines

Sincerely,
Jing Han
Editor
Microbiology Spectrum

Reviewer #1 (Comments for the Author):

The authors provided a detailed response to the issues raised by the reviewers, and additional data is provided. However, two outstanding question remain:

1. The authors still do not include any statistical analysis for Figs 2 and 4 as indicated by one of the reviewers. This should be

added. For figure, they have the quantitative data available (fluorescence intensity -median and SD-) and they do indicate that the experiments were done three independent times. So there is no barrier to run a one WAY ANOVA analysis with multiple comparisons correction.

2. Authors still have not addressed experimentally the possibility that the toxin is secreted by the T6SS but it is NOT injected into the prey cells by the T6SS. This is one of the key points of the manuscript, The two experiments suggested previously (comment 3) can be easily done by the authors with the tools they have using very simple assays.

Response to reviewers

We appreciate reviewer 1 positive feedback and valuable suggestions, which have significantly improved our revised manuscript. Here, we provide a detailed response to reviewer 1 comments and the corresponding changes made to the revised manuscript. Please note the following: Text in black font corresponds to the reviewers' comments, blue font text is our response to the reviewers' comments, and highlighted text represents the excerpts from the revised manuscript.

Response to Reviewer 1

The authors provided a detailed response to the issues raised by the reviewers, and additional data is provided. However, two outstanding question remain:

Re: We greatly appreciate the reviewer 1's invaluable comments and suggestions on our work, which are extremely helpful for us to improve our manuscript.

Comment 1: The authors still do not include any statistical analysis from Figs 2 and 4 as indicated by one of the reviewers. This should be added. For figure, they have the quantitative data available (fluorescence intensity -median and SD-) and they do indicate that the experiments were done three independent times. So there is no barrier to run a one WAY ANOVA analysis with multiple comparisons correction.

Re: We fully agree with the reviewer.

1) For the bacterial growth experiment shown in Fig. 2, we replotted as suggested by reviewer 1 and run a one-way ANOVA analysis with multiple comparisons test. The updated Fig. 2 is as follows.

Figure 2. YPK_0952 is toxic in *Y. pseudotuberculosis* and *E. coli*. (A) Growth curves of *Y. pseudotuberculosis* harboring indicated plasmids were obtained by measuring OD₆₀₀ at 2 h intervals. Ordinary one-way ANOVA with Tukey's multiple comparison test with $\Delta ypk_0952\Delta ypk_0953$ (*ypk_0952*). (B) Growth of *Y. pseudotuberculosis* harboring indicated plasmids on YLB solid medium. From left to right are increasing serial 10-fold dilutions. (C, E) Growth curves of *E. coli* BL21 (DE3) harboring indicated plasmids were obtained by measuring OD₆₀₀ at 2 h intervals. Ordinary one-way ANOVA with Tukey's multiple comparison test with *ypk_0952*. (D, F) Growth of *E. coli* BL21 (DE3) harboring plasmids with the indicated genes, or empty plasmids, under inducing (IPTG) conditions. From left to right are increasing serial 10-fold dilutions. Ordinary one-way ANOVA with Tukey's multiple comparison test with Vector. Error bars represent the mean \pm standard deviation (SD) of three independent experiments. * $P < 0.0332$; ** $P < 0.0021$; *** $P < 0.0002$, **** $P < 0.0001$; ns, not significant.

2) We processed the fluorescence data in Fig. 4 using FlowJo software, calculated the median and SD of the fluorescence intensity, and performed a one-way ANOVA with Tukey's multiple comparison test. Please see the revised Supplementary Figure 5 below.

Supplementary Figure 5: Flow cytometry experiment median TUNEL FITC. (A) Flow cytometry analysis of the median TUNEL FITC of DNA fragments before IPTG induction. (B) Flow cytometry analysis of the median TUNEL FITC of DNA fragments after 4 hours of IPTG induction. Error bars represent the mean \pm SD of three independent experiments, ordinary one-way ANOVA with Tukey's multiple comparison test with Vector. * $P < 0.0332$; ** $P < 0.0021$; *** $P < 0.0002$, **** $P < 0.0001$; ns, not significant.

Comment 2: Authors still have not addressed experimentally the possibility that the toxin is secreted by the T6SS but it is NOT injected into the prey cells by the T6SS. This is one of the key points of the manuscript, The two experiments suggested previously (comment 3) can be easily done by the authors with the tools they have using very simple assays.

Re: We fully agree with the reviewer.

1) Following the method described by Song et al, we conducted intra-species and inter-species contact-independent competition experiments using a membrane to separate predators and prey to assess killing (Song et al. *Nat. Commun.* 2021, 12,

423-434). As shown in Supplementary Figure 7A, the predator (WT) displayed a growth advantage against prey ($\Delta ypk_0952\Delta ypk_0953$), but this advantage disappeared when *ypk_0953* was introduced into the prey. Additionally, results in Supplementary Figure 7B demonstrate that predator WT significantly inhibited *E. coli* DH5 α compared to the Δypk_0952 mutant. This indicates that the YPK_0952 toxin can be secreted into the supernatant via T6SS-3 and internalized into prey cells to complete toxin delivery. The experimental details were described in the revised manuscript, page 13 “For intra-species contact-independent competition performed on a solid surface, 5 μ L of prey strain was first spotted onto a 0.22 μ m nitrocellulose membrane on M9 agar plates. After the liquid dried, another layer of 0.22 μ m nitrocellulose membrane was placed on top, and 5 μ L of predator strain was spotted at the same position on the second membrane. The plates were then incubated at 26 $^{\circ}$ C for 48 hours” and “The same method as used for intra-species competition was employed to conduct contact-independent competition on a solid surface, 5 μ L of prey strain was first spotted onto a 0.22 μ m nitrocellulose membrane on M9 agar plates. After the liquid dried, another layer of 0.22 μ m nitrocellulose membrane was placed on top, and 5 μ L of predator strain was spotted at the same position on the second membrane. The plates were then incubated at 26 $^{\circ}$ C for 24 hours”. The corresponding description was added in the revised manuscript, page 17 “Further analysis of contact-independent competition experiments revealed that YPK_0952 could still mediate the bactericidal function of T6SS when prey and predator bacteria were isolated using a 0.22 μ m nitrocellulose membrane (Supplementary Fig. 7A). It was shown that *Y. pseudotuberculosis* T6SS-3 utilizes YPK_0952 for both contact-dependent and contact-independent bacterial competition” and “To further investigation of YPK_0952 mediated contact-dependent and contact-independent T6SS killing across different species, we performed competition assays on solid media. The WT strain exerted a robust inhibitory effect on the growth of *E. coli* DH5 α , while the Δypk_0952 mutant faltered in inhibiting the growth of *E. coli* DH5 α . Notably, the diminished sensitivity of the Δypk_0952 mutant was substantially reversed through complementation (Fig. 5B and Supplementary Fig. 7B)”. The updated

Supplementary Figure 7 is as follows.

Supplementary Figure 7: YPK_0952 mediates contact-independent T6SS killing.

(A) Contact-independent intra-species growth competition experiments performed by physically separating the indicated *Y. pseudotuberculosis* predator and prey strains with a membrane, and culturing them on the surface of solid medium at 26 °C for 48 hours. The CFU of the prey strains was measured based on plate counts. (B) Contact-independent Inter-species growth competition experiments performed by physically separating the indicated *Y. pseudotuberculosis* predator and *E. coli* DH5α prey strains with a membrane, and culturing them on the surface of solid medium at 26 °C for 24 hours. The CFU of the prey strains were measured based on plate counts. Error bars represent the mean ± SD of prey CFU from three independent experiments, with two-tailed, unpaired Student's *t*-test. **P* < 0.0332; ***P* < 0.0021; ****P* < 0.0002, *****P* < 0.0001; ns, not significant.

2) To test the DNA degradation ability and antibacterial performance of WT, T6SS-3 null strain, and effector mutants, we conducted DNase activity in vitro, DAPI staining, and antibacterial experiments following the research methods of Pissaridou and Song et al (Pissaridou et al. *Proc. Natl. Acad. Sci. U.S.A.* 2018, 115, 12519-12524; Song et al. *Nat. Commun.* 2021, 12, 423-434).

1. Induce the secretion of toxins from WT, ΔclpV3, and Δyjk_0952 strains in M9 media, and then the obtained toxin-containing sterile supernatant was incubated with

λ DNA. As shown in Supplementary Figure 6A, after 2 hours of incubation, the supernatant from WT degraded λ DNA most effectively, while the supernatants from $\Delta clpV3$ and Δypk_0952 showed almost no degradation of λ DNA.

2. The aforementioned toxin-containing sterile supernatants were co-incubated with *E. coli* DH5 α for 4 hours, after which the bacteria were collected for DAPI staining and observed under fluorescence confocal microscopy. As shown in Supplementary Figure 6B, the supernatant from WT significantly degraded the DNA of DH5 α , whereas there was no noticeable difference in the DAPI fluorescence of DH5 α treated with the supernatants from $\Delta clpV3$ and Δypk_0952 .

3. Further, the supernatants after co-incubation with DH5 α for 4 hours were serially diluted and plated to count the CFU of DH5 α , to test the antibacterial performance of the WT, $\Delta clpV3$, and Δypk_0952 . As shown in Supplementary Figure 8, the CFU of DH5 α in the WT supernatant were significantly lower than those in $\Delta clpV3$ and Δypk_0952 . In conclusion, the YPK_0952 toxin relies on T6SS-3 for secretion into the extracellular space, after which it internalizes into prey cells to exert DNase activity and gain a competitive advantage.

We carefully analyzed the experimental results and added new discussions to the manuscript. The corresponding description was added in the revised manuscript, page 21 “S-type pyocin loci typically encode two protein components: a high molecular weight effector protein and a low molecular weight immunity protein (57). The effector protein usually consists of three to four domains, with the N-terminus domain involved in recognition of cell surface receptors, the function of the second domain currently unclear, the third domain responsible for pyocin translocation and penetration, and the C-terminus domain possessing lethal activity (43, 44). Like most bacteriocins, S-type pyocins bind to specific receptors on the outer membrane of closely related bacterial species, then translocate across the membrane and kill their targets. Studies indicate that most S-type pyocins bind to TonB-dependent outer membrane receptors on bacterial cells. For example, pyocins S2 and S4 enter cells through the FpvAI protein (57); pyocin S3 targets FpvAII (58); while pyocin S5 recognizes the FptA ferripyochelin receptor (59). This study found that Y.

pseudotuberculosis T6SS-3 gains a contact-independent competitive advantage by secreting the pyocin-like effector, YPK_0952, into the extracellular space, with the T6SS mechanism acting as a conduit for delivering effectors to the outer membrane receptors of prey cells. However, the internalization mechanism of YPK_0952 remains unclear, whether it resembles the cell membrane receptor-mediated mechanism of pyocins, will be further investigated in future studies”.

The updated Supplementary Figure 6 and Supplementary Figure 8 are as follows.

Supplementary Figure 6: The ability of WT, $\Delta clpV3$, and Δypk_0952 to degrade DNA. (A) Capacity of WT, $\Delta clpV3$, and Δypk_0952 sterile supernatants to degrade DNA in vitro. The sterile supernatants (13 μ L) of WT, $\Delta clpV3$, and Δypk_0952 strains were incubated with λ DNA (0.35 μ g). The products were analyzed using agarose gel. All gel results were independently repeated three times with similar outcomes. (B) YPK_0952 functions as DNase in vivo. The sterile supernatants (5 mL) of WT, $\Delta clpV3$, and Δypk_0952 strains were incubated with DH5 α and the activity of YPK_0952 in the cells was observed using fluorescence microscopy, M9 medium as a negative control. All micrographs were independently repeated three times with similar results. DAPI, fluorescence observation; BF, bright field. Scale bar: 10 μ m.

Supplementary Figure 8: The antibacterial properties of WT, $\Delta clpV3$, and Δypk_0952 . The sterile supernatants (5 mL) of WT, $\Delta clpV3$, and Δypk_0952 strains were co-incubated with DH5 α at 37 °C for 4 hours, followed by serial dilution of the cultures and plating on LB agar plates. The CFU of DH5 α strains were measured based on plate counts. Error bars represent the mean \pm SD of DH5 α CFU from three independent experiments, with two-tailed, unpaired Student's *t*-test. **P* < 0.0332; ***P* < 0.0021; ****P* < 0.0002, *****P* < 0.0001; ns, not significant.

References

1. Song L, Pan J, Yang Y, Zhang Z, Cui R, Jia S, Wang Z, Yang C, Xu L, Dong TG, Wang Y, Shen X. 2021. Contact-independent killing mediated by a T6SS effector with intrinsic cell-entry properties. *Nat. Commun.* 12:423-434.
2. Pissaridou P, Allsopp LP, Wettstadt S, Howard SA, Mavridou DAI, Filloux A. 2018. The *Pseudomonas aeruginosa* T6SS-VgrG1b spike is topped by a PAAR protein eliciting DNA damage to bacterial competitors. *Proc. Natl. Acad. Sci. U.S.A.* 115:12519-12524.

Re: Spectrum04278-23R2 (A pyocin-like T6SS effector mediates bacterial competition in *Yersinia pseudotuberculosis*)

Dear Prof. Yao Wang:

Your manuscript has been accepted, and I am forwarding it to the ASM production staff for publication. Your paper will first be checked to make sure all elements meet the technical requirements. ASM staff will contact you if anything needs to be revised before copyediting and production can begin. Otherwise, you will be notified when your proofs are ready to be viewed.

Sincerely,
Jing Han
Editor
Microbiology Spectrum

Reviewer #1 (Comments for the Author):

The authors have done a good job addressing the final comments made.